

# DINOSTRAT: A global database of the stratigraphic and paleolatitudinal distribution of Mesozoic-Cenozoic organic-walled dinoflagellate cysts

Peter K. Bijl

Department of Earth Sciences, Utrecht University, Utrecht, 3584 CB, the Netherlands

*Correspondence to*: Peter K. Bijl (p.k.bijl@uu.nl)

**Abstract.** Mesozoic–Cenozoic organic-walled dinoflagellate cyst (dinocyst) biostratigraphy is a crucial tool for relative and absolute age control in complex ancient sedimentary systems. However, stratigraphic ranges of dinocysts are found to be strongly diachronous geographically. A global compilation of state-of-the-art calibrated regional stratigraphic ranges could assist in quantifying regional differences and evaluate underlying causes. For this reason, DINOSTRAT is here initiated – an open source, iterative, community-fed database intended to house all regional chronostratigraphic calibrations of dinocyst events (https://github.com/bijlpeter83/DINOSTRAT.git). DINOSTRAT version 1.0 includes >8500 entries of first and last occurrences (collectively called "events") of >1900 dinocyst taxa, and their absolute ties to the chronostratigraphic time scale of Gradstein et al., 2012. Entries are derived from 199 publications and 189 sedimentary sections. DINOSTRAT interpolates paleolatitudes of regional dinocyst events, allowing evaluation of the paleolatitudinal variability of dinocyst event ages. DINOSTRAT allows for open accessibility and searchability, on region, age, and taxon. This paper presents a selection of the data in DINOSTRAT: (1) the (paleo)latitudinal spread and evolutionary history of modern dinocyst species; (2) the evolutionary patterns and paleolatitudinal spread of dinoflagellate cyst (sub)families; (3) a selection of key dinocyst events which are particularly synchronous. Although several dinocysts show – at the resolution of their calibration – quasi-synchronous event ages, indeed many species have remarkable diachroneity. DINOSTRAT provides the data storage approach by which the community can now start to relate diachroneity to (1) inadequate tie to chronostratigraphic time scales; (2) complications in taxonomic concepts and (3) ocean connectivity and/or the affinities of taxa to environmental conditions.

## 1 Introduction

Over 50 years of research efforts have established a framework to use organic-walled dinoflagellate cysts (dinocysts) as biostratigraphic and chronostratigraphic tool. Dinocyst biostratigraphy is particularly applied to sediments which are difficult to date otherwise, such as restricted/nearshore marine settings (e.g., Poulsen et al., 1994; Brinkhuis et al., 1998; Iakovleva et

al., 2001; Śliwińska et al., 2012; Clyde et al., 2014), and polar regions (e.g., Sluijs et al., 2006; Bijl et al., 2013a; Houben et al., 2013; Radmacher et al., 2015; Śliwińska et al., 2020). As with all biostratigraphy, the reliability of dinocyst biostratigraphy



heavily depends on the accuracy, precision, and regional consistency of the absolute ages of first and last stratigraphic occurrences (hereafter jointly referred to as "events") of easily recognized taxa. Through the past decades, absolute ages of dinocyst events were increasingly better chronostratigraphically constrained, using independent age control from

magnetostratigraphy (e.g., Brinkhuis et al., 1992; Powell et al., 1996), other biostratigraphic tools (e.g., Davey, 1979; Leereveld, 1997; Oosting et al., 2006; Awad and Oboh-Ikenobe, 2019), and astrochronology (Versteegh, 1997). However, efforts to globally compile the chronostratigraphic calibration of dinocyst events revealed strong diachroneity for many species, between broad latitudinal bands, and endemism of many species within latitudinal bands (e.g., Williams et al., 2004). Because this impacts the development of quasi-global dinocyst zonation schemes, as has been proposed for other microfossil groups

(e.g., Martini, 1971; Gradstein et al., 2020), the question is how the research field of dinocyst biostratigraphy should progress. Two questions arise from the notion of geographic diachroneity of dinocyst events:

- What kind of error or uncertainty should be applied to the absolute ages of events? Now that diachroneity has been demonstrated, the next step is to quantify the uncertainty in absolute ages of dinocyst events for each species, and to assess regional consistency. This is particularly important when calibrated species ranges are geographically

extrapolated over large distances. And a related question: What is the impact of regional variability in absolute ages of events on the regional consistency of the stratigraphic order of events?

- What are the underlying causes for the observed diachroniety? Broadly, 3 reasons could apply: (1) inaccurate or inadequate tie of dinocyst events to the chronostratigraphic time scale, which leads to apparent (but perhaps false) diachroniety of species events between sites; (2) complexities in taxonomic concepts could obscure comparison of

species ranges between sites. This aspect relates to the ease by which subtle morphological differences between species can be recognized (e.g., Hoyle et al., 2019). It also relates to the question whether the last occurrence of a fossil dinocyst taxon reflects extinction of its producer, adjustment of cyst morphology by its producer (e.g., Rochon et a., 2009), or a change in its life cycle strategy (e.g., towards less-preservable pellicle cysts; Bravo and Figueroa, 2014); (3) finally, paleoenvironmental/paleoceanographic conditions can impact species occurrence: ocean

connectivity (Van Simaeys et al., 2005; Bijl et al., 2013b; Van Helmond et al., 2016), leads and lags in the biotic response to climate change (e.g., Sluijs et al., 2007) or the temperature affinity of dinocyst taxa (Van Simaeys et al., 2005; Van Helmond et al., 2016). For instance, in geologic time intervals of global climate cooling, warm-loving plankton species have diachronous last occurrences (LOs) which are progressively later at lower latitudes. A good example is the modern occurrence in the western Pacific warm pool of *Dapsilidinium pastielsii*, a species that was

long thought to be extinct in the Pliocene (Head et al., 1989). This is exemplary for how important it is to assume asynchronous biostratigraphic events as potential paleoceanographic signal, or a signal of paleoecologic affinity, rather than biostratigraphic error.

A process towards answering these, and improving the accuracy of dinocyst biostratigraphy, requires a data compilation

approach that houses data from as many sites as possible, with detailed metadata on paleogeographic evolution of sites, and



the means of chronostratigraphic calibration. It further requires that such data compilations are constantly updated to new insights: updated geologic time scales and bio- magnetostratigraphic zonation schemes, altered taxonomic concepts, new age models of sections, new stratigraphic sections. A complication on a logistical front, is that dinocyst ranges are typically published in the closed-access peer-review literature, which are not easily accessible to all, are inconsistent in their approach,

and not easily updated to new insights.

This paper initiates DINOSTRAT, an open-source, online platform intended to house, disseminate, and iteratively update all published chronostratigraphic calibrations of dinoflagellate cyst ranges: the way in which they are tied to the chronostratigraphic time scale and the (paleo-) geographic position of the site from which they were calibrated. DINOSTRAT version 1.0 currently contains over 8500 entries of first and last occurrences of over 1900 dinoflagellate cyst taxa to the

international time scale. These entries originate from 199 peer-reviewed papers presenting data from 189 sites. Including as many reports/sites as possible, with verifiable independent age control, and their latitudinal evolution through time, allows for proper evaluation of error and uncertainty. DINOSTRAT will allow to assess and quantify regional variability/consistency in event ages and provides the basic information to evaluate the paleoceanographic signal that diachroniety may hold. Open accessibility of the basic dinocyst stratigraphic data would further allow a proper evaluation and update of evolutionary patterns

in dinocyst families (McRae et al., 1996) with full disclosure of the underlying data. The approach on paper selection, data entry and calculations of ages and paleolatitudes is explained in Section 2. Section 3 presents examples of calibrated dinoflagellate cyst events: the stratigraphic and paleolatitudinal distribution of selected modern dinoflagellate cysts, and that of extant and extinct dinocyst families, with selected taxa highlighted. Section 4 discusses the implications of the DINOSTRAT approach and further directions. This paper represents the start of a community-fed data assembly approach to iteratively

improve regional constraints on dinoflagellate cyst biostratigraphy.

## 2 Approach

DINOSTRAT version 1.0 represents a compilation of dinocyst events from peer-reviewed literature, with a publication date predating January 1st, 2021 (see Table 1). The taxonomic nomenclature, supra-generic classification and synonymy cited in Williams et al. (2017) is followed. One inherent assumption in the initial setup of DINOSTRAT is that the authors of the

reviewed literature have applied a consistent taxonomic framework. DINOSTRAT reports events of dinocyst species as they were presented in the papers, but applying the synonymy index of Williams et al., 2017. Most dinoflagellate cyst species are easily recognized, have a stable morphology (both regionally and through time) and clearly defined species concepts. However, some species (and subspecies) diagnoses are more subtle or represent endmembers in a continuum (e.g., Hoyle et al., 2019), in part imposed by the environmental conditions (e.g., Ellegaard, 2000). Some authors tend to lump species in complexes,

while others split into subspecies. The international recognition of these lumps and splits may have evolved through time, and may have restricted, regional significance only. Therefore, subtle differences in species concept interpretation may exist



between authors and regions, which the current approach was unable to account for, and is considered a next step, when individual studies or sites are revisited.

For the subfamily Wetzelielloideae, DINOSTRAT deviates from the taxonomic index of Williams et al. (2017). The
fundamental redefinition of species concepts in the taxonomic revisions for the Wetzelielloideae (Williams et al., 2015) eliminates many stratigraphically useful Eocene dinocyst taxa (Bijl et al., 2016). Therefore, for this subfamily, the calibration of dinocyst species is presented in the taxonomic classification of Wetzelielloideae prior to (Williams et al., 2015).

**Table 1: Papers used in this review. Reference, Geography, Age base and Age top (in Ma), Tier (see Figure 1) and**
**means of calibration to the Geologic Time Scale (GTS).**

| Reference | Geography (location) | Age base | Age top | Tier | Calibrated to |
|---|---|---|---|---|---|
| Açikalin et al., 2015 | NW Turkey | 67 | 65 | 3 | planktonic foraminifera stratigraphy on the same section |
| Århus et al., 1989 | Norway | 166 | 155 | 3 | Russian Platform zones, converted to Boreal ammonite zones (see Supplement 1) |
| Aubry et al., 2020 | Labrador Sea, Greenland | 3.2 | 2.25 | 1 | Magnetostratigraphy on the same samples. Magnetic reversals were calibrated usingplanktonic foraminifera and nannofossils |
| Awad and Oboh-Ikenobe, 2016 | Ivory Coast Margin | 57 | 54 | 3 | CP nannoplankton stratigraphy on the same section |
| Awad and Oboh-Ikenobe, 2019 | Ivory Coast Margin | 28 | 16 | 3 | CP/CN nannofossil stratigraphy on the same samples |
| Bailey et al., 1997 | UK | 157 | 152 | 3 | Boreal ammonite stratigraphy from the same core samples. |
| Baruffini et al., 2002 | S Italy | 35 | 32 | 3 | CP nannoplankton stratigraphy from the same study |
| Besems, 1992 | Borneo | 65 | 0 | 5 | chronostratigraphy, no independent age controll shown (Industry data) |
| Biffi and Manum, 1988 | Central Italy | 36 | 22 | 3 | NP/NN nannoplankton and N/P planktonic foraminifer stratigraphies from the same sections |
| Bijl and Brinkhuis, 2015 | East Antarctica | 54 | 47 | 2 | Magnetostratigraphy on the same section. Magnetochrons are dated using dinocyst biostratigraphy |





| Bijl et al 2013, 2014 | SW Pacific | 57 | 35 | 2 | Complicated paleomagnetic signal and isotope stratigraphic constraints at Site 1171 and 1172. U1356 was calibrated to magnetostratigraphy, using dinocyst biostratigrapy |
|---|---|---|---|---|---|
| Bijl et al., 2018 | East Antarctica | 34 | 10 | 1 | Magnetostratigraphy with nannoplankton stratigraphy |
| Bowman et al., 2012 | Seymour Island, Antarctica | 68 | 65 | 4 | Inferred position of the K/Pg Boundary |
| Bowman et al., 2016 | Seymour Island, Antarctica | 66 | 57 | 4 | Inferred position of the K/Pg Boundary |
| Brinkhuis, 1994 | Italy | 35 | 33 | 1 | Magnetostratigraphy with NP/CP nannoplankton and foraminifer stratigraphy in the same sections |
| Brinkhuis and Biffi, 1993 | Central Italy | 37 | 32 | 1 | Magnetostratigraphy, based on nannoplankton stratigraphy and foraminifer stratigraphy |
| Brinkhuis et al., 1992 | NW Italy | 26 | 22 | 1 | Magnetostratigraphy, interpreted based on nannoplankton stratigraphy and foraminifer stratigraphy |
| Brinkhuis et al., 1998 | Tunesia, Denmark | 67 | 65 | 3 | Planktonic foraminifer stratigraphy at the same sections |
| Brinkhuis et al., 2003a | Western Tasmania | 36 | 1 | 2 | Magnetostratigraphy with sparse nannoplankton in the Eocene. Oligocene and Neogene calibrated to nannoplankton-, foraminifer- and magnetostratigraphy |
| Brinkhuis et al., 2003b | Eastern Tasmania | 70 | 30 | 2 | A complicated paleomagnetic signal with isotope stratigraphic constraints (see Dallanave et al., 2016). |
| Brown and Downie, 1984 | Rockall Plateau, Ireland | 58 | 33 | 3 | CNP nannoplankton stratigraphy on the same cores |
| Brown and Downie, 1985 | Northern Bay of Biscay, France | 60 | 10 | 3 | NP/NN nannoplankton stratigraphy on the same cores |
| Bucefalo Palliani and Riding, 1997a | Italy | 183 | 179 | 4 | Tethyan ammonite stratigraphy, but no ammonite data was shown |
| Bucefalo Palliani and Riding, 1997b | France | 199 | 170 | 4 | Boreal ammonite stratigraphy, but no ammonite data was shown (see conversions in Supplement 1) |





| | | | | | |
|---|---|---|---|---|---|
| Bucefalo Palliani and Riding, 2000 | UK | 200 | 179 | 4 | Tethyan ammonite stratigraphy, but no ammonite data was shown (see conversions in Supplement 1) |
| Bucefalo Palliani and Riding, 2003 | Boreal/Tethys | 191 | 180 | 4 | Boreal ammonite stratigraphy, but no ammonite data was shown. |
| Bujak and Matsuoka, 1986 | North Pacific, Japan | 23 | 0 | 5 | Independent age constraints from planktonic foraminifera, radiolaria, diatoms and nannoplankton are not shown in the paper. |
| Bujak and Mudge, 1994 | North Sea, UK | 57 | 53 | 4 | Synthesis. Plots dinocyst events against NP nannoplankton and P planktonic foraminifer stratigraphy not presenting independent stratigraphic data. |
| Correia et al., 2019 | Portugal | 183 | 168 | 3 | Tethyan ammonite stratigraphy on the same sections. |
| Costa and Davey, 1992 | North Sea, UK | 145 | 66 | 4 | Ammonite zones are plotted but no ammonite data was presented. Campanian-Maastrichtian events were calibrated to stages (see conversions in Supplement 1) |
| Costa and Downie, 1979 | N Atlantic | 58 | 5 | 3 | Nannoplankton stratigraphy on the same section |
| Crouch et al., 2014 | New Zealand | 66 | 54 | 1 | Magnetostratigraphy and NP nannoplankton stratigraphy on the same samples |
| Dallanave et al., 2016; Crouch et al., 2020 | New Zealand | 54 | 46 | 1 | Magnetostratigraphy and NP nannoplankton zones on the same section |
| Davey, 1979 | N atlantic | 124 | 100 | 3 | Nannoplankton stratigraphy on the same section |
| Davey, 1982 | Denmark | 152 | 125 | 3 | Original stratigraphic account was based on Ammonites, pelycepods and benthic foraminifera (see conversions in Supplement 1) |
| Davey, 2001 | UK | 134 | 131 | 3 | Boreal Ammonite stratigraphy on the section |
| Davey and Verdier, 1971 | France | 113 | 103 | 4 | Boreal ammonite stratigraphy, not shown (see conversions in Supplement 1) |
| De Lira Mota et al., 2020 | Gulf Coast, USA | 37 | 32 | 3 | NP nannofossil stratigraphy on the same samples |
| De Schepper and Head 2008, 2009 | North Atlantic | 6 | 0 | 1 | Magnetostratigraphy, NN nannofossil stratigraphy and N planktonic foraminifer stratigraphy |



| | | | | | |
|---|---|---|---|---|---|
| De Schepper et al., 2017 | North Atlantic | 7 | 1 | 1 | Magnetostratigraphy on the same section |
| De Vernal and Mudie, 1989 | Labrador Sea, Greenland | 5.5 | 0 | 3 | Shipboard NN nannofossil stratigraphy |
| De Vernal et al., 1992 | North Atlantic | 1.5 | 0 | 1 | Magnetostratigraphy and NN nannofossil stratigraphy |
| De Verteuil and Norris, 1996 | Chesapeake Bay, USA | 25 | 4 | 4 | Synthesized stratigraphic data, no independent age control presented |
| Dimter and Smelror, 1990 | sw Germany | 166 | 163 | 3 | Boreal ammonite zonation on the same material |
| Dodsworth, 2000 | USA and UK | 96 | 93 | 3 | Planktonic foraminifer and ammonite stratigraphy on the same section |
| Duffield and Stein, 1986 | Gulf Coast, USA | 35 | 5 | 3 | N Planktonic foraminiferal zonation |
| Duque-Herrera et al., 2018 | Colombia | 18 | 5 | 3 | NN nannofossil events in the same core |
| Duxbury, 1983 | North Sea | 126 | 110 | 3 | Boreal ammonite zonation (see conversions in Supplement 1) |
| Duxbury, 2001 | Scotland | 139 | 100 | 4 | Boreal ammonite zonation, not directly from the well cutting material (see conversions in Supplement 1) |
| Dybkjær and Piasecki, 2008, 2010 | Denmark | 23 | 0 | 3 | NP/NN nannoplankton stratigraphy |
| Egger et al., 2016 | Newfoundland, USA | 35 | 21 | 1 | Magnetostratigraphy with NN nannoplankton stratigraphy |
| Eldrett and Harding, 2009 | Voring Plateau, Norwegian Sea | 52 | 33 | 2 | Magnetostratigraphy on the same section, see Eldrett et al., 2004 |
| Eldrett et al., 2004 | Norwegian Sea | 50 | 30 | 2 | Magnetostratigraphy, but chrons were not independently interpreted |
| Eldrett et al., 2019 | North Atlantic | 34 | 24 | 2 | Magnetostratigraphy on the same section, see Eldrett et al., 2004 |
| Eshet et al., 1992 | Israel | 67 | 65 | 3 | NP nannoplankton strat at the same site |
| Feist-Burkhardt and Monteil, 1997 | France | 171 | 167 | 3 | Calibrated to Boreal ammonite stratigraphy (see conversions in Supplement 1) |
| Feist-Burkhardt, 1990 | sw Germany | 174 | 168 | 3 | Boreal ammonite stratigraphy |



| Fensome et al., 2008 | Scotian Margin, E Canada | 100 | 0 | 3 | NN and NC nannoplankton stratigraphy, but because based on cuttings, only LADs are given |
|---|---|---|---|---|---|
| Firth et al., 2013 | North Atlantic | 51 | 32 | 1 | Magnetostratigraphy with independent age control from nannoplankton and planktonic foraminifer stratigraphy |
| Firth, 1996 | N Atlantic | 45 | 30 | 1 | Calibrated using magnetostratigraphy from Eldrett et al., 2009 |
| Frieling et al., 2014 | West Siberian Sea | 60 | 45 | 2 | Magnetostratigraphy and stable carbon isotope stratigraphy |
| Gradstein et al., 1992 | North Sea, NL | 66 | 23 | 3 | N/P foraminifer stratigraphy, but entered against NP nannoplankton stratigraphy |
| Grothe et al., 2017 | Black Sea | 6 | 5.5 | 1 | Magnetostratigraphy on the same section |
| Guasti et al., 2005 | Tunisia | 66 | 57 | 3 | NP nannoplankton and P foraminifer stratigraphy on the same section |
| Habib and Drugg, 1983 | East Coast USA | 157 | 138 | 1 | Magnetostratigraphy on the same section |
| Habib and Drugg, 1987 | East Coast USA | 145 | 66 | 2 | Magnetostratigraphy on the same section |
| Harding et al., 2011 | S Russia | 152 | 134 | 3 | Russian ammonite zonation on the same sections, correlated to Boreal ammonite zones (see conversions in Supplement 1) |
| Harland, 1979 | N atlantic | 12 | 0 | 3 | Nannoplankton stratigraphy on the same section |
| Harland, 1992 | North Sea | 23 | 0 | 4 | NN nannoplankton and N planktonic foraminifer stratigraphy, but independent age constraints not explicitly shown |
| Head, 1998 | North Sea | 4 | 1.6 | 4 | Stages, using known ages of sampled formations |
| Head and Norris, 1989 | Western North Atlantic | 57 | 23 | 3 | NN nannoplankton stratigraphy |
| Head and Norris, 2003 | North Atlantic | 7 | 1 | 1 | Magnetostrat and NC nannoplankton stratigraphy from the same section |
| Head et al., 1989 | Labrador Sea | 9 | 5 | 3 | NN and CN nannoplankton stratigraphy at the same site |





| Heilmann-Clausen, 1985 | North Sea | 62 | 54 | 3 | NP nannofossil zones on the same section |
|---|---|---|---|---|---|
| Heilmann-Clausen, 1987 | Danish basin | 152 | 100 | 4 | Synthesis of records from the North Sea area. Correlation to Boreal ammonite zones (see conversions in Supplement 1) |
| Heilmann-Clausen and Van Simaeys, 2005 | Danish North Sea | 48 | 30 | 3 | NP nannofossil zonation |
| Helby and McMinn, 1992 | NW Australia | 139 | 104 | 3 | CC nannofossil zonation on the same section |
| Helby et al., 1987 | Australia | 241 | 66 | 4 | Synthesis, calibrated to stages using industry information. Albian-Danian has independent age controll from foraminiferal and nannoplankton zones |
| Hoek et al., 1996 | Israel | 73 | 69 | 3 | CC and UC nannofossil events |
| Hollis et al., 2009 | New Zealand | 51 | 46 | 3 | NP nannofossil stratigraphy on the same section |
| Houben et al., 2011 | Falkland Islands, S Atlantic | 35 | 32 | 1 | Oi-1 isotope event, the age of which is then transferred to the GPTS |
| Houben et al., 2019 | Alabama, USA | 37 | 30 | 1 | Magnetostratigraphy and NP nannoplankton stratigraphy on the same section |
| Iakovleva and Heilmann-Clausen, 2010 | Siberia | 52 | 35 | 2 | Magnetostratigraphy on the same section |
| Ioannides et al., 1988 | France | 157 | 152 | 3 | Boreal ammonite stratigraphy (see conversions in Supplement 1) |
| King et al., 2018 | Crimea | 59 | 48 | 3 | NP nannofossil stratigraphy on the same samples |
| Kirsch, 1991 | Bad Tolz, Southern Germany | 94 | 66 | 3 | Planktonic foraminifer stratigraphy, data not shown |
| Köthe, 2012 | NW Germany | 65 | 0 | 3 | NP nannoplankton stratigraphy in the same sections. (for conversions see Supplement 1) |
| Köthe et al., 1988 | Pakistan | 58 | 50 | 3 | Nannoplankton stratigraphy on the same sections |
| Krijgsman et al., 1995 | Mediterranean (Gibliscemi) | 10 | 7 | 1 | Magnetostratigraphy with planktonic foraminifer stratigraphy on the same section |





| Kuhlman et al., 2006 | Central North Sea | 4 | 0 | 1 | Magnetostratigraphy with foraminifer stratigraphy on the same section |
|---|---|---|---|---|---|
| Lebedeva et al., 2013 | Omsk, sw siberia | 83 | 68 | 1 | Magnetostratigraphy and CC nannoplankton stratigraphy on the same section |
| Leereveld, 1995 | Caravaca, Southern Spain | 145 | 105 | 3 | Tethyan ammonite stratigraphy on the same section (for conversions see Supplement 1) |
| Leereveld, 1997a | Caravaca, Southern Spain | 134 | 125 | 3 | Tethyan ammonite stratigraphy on the same section (for conversions see Supplement 1) |
| Leereveld, 1997b | Caravaca, Southern Spain | 146 | 134 | 3 | Tethyan ammonite stratigraphy on the same section |
| Londeix and Jan Du Chene, 1998 | Bordeaux, France | 21 | 16 | 3 | NN nannoplankton stratigraphy |
| Louwye et al., 2004 | Belgium | 6 | 0 | 3 | NN nannoplankton stratigraphy on the same section |
| Louwye et al., 2008 | Porcupine basin, Ireland | 17 | 11 | 1 | Magnetostratigraphy on the same section |
| Mao and Mohr, 1992 | Kerguelen Plateau, Antarctica | 75 | 70 | 3 | CC nannofossil stratigraphy on the same section |
| Marret et al., 2020 | global | 0 | 0 | | Surface sediment data |
| Masure, 1988 | Ivory Coast Margin | 140 | 112 | 3 | CC nannofossil stratigraphy on the same section |
| Masure et al., 1998 | Ivory Coast Margin | 90 | 57 | 3 | CP and CC nannoplaknton stratigraphy on the same section |
| Matsuoka et al., 1987 | Japan | 20 | 0 | 3 | N foraminifer events on the same section |
| Matthiessen and Brenner, 1996 | Spitsbergen | 3 | 0 | 1 | Magnetostratigraphy on the same section |
| McLachlan et al., 2018 | western Canada | 77 | 71 | 1 | Magnetostratigraphy on the same site |
| McMinn, 1992 | NW Australia | 16 | 3 | 3 | CP nannofossil stratigraphy and N planktonic foraminifer stratigraphy |
| McMinn, 1993 | NW Australia | 9 | 0 | 3 | CN nannoplankton stratigraphy on the same section |
| Mohr and Mao, 1997 | Kerguelen and Maud Rise, Antarctica | 73 | 70 | 1 | Magnetostratigraphy, CC nannoplankton stratigraphy |



| Montanari et al., 1997 | Contessa, Gubbio, Italy | 26 | 16 | 1 | Magnetostratigraphy, foraminifer and nannoplankton stratigraphy |
|---|---|---|---|---|---|
| Monteil, 1992 | France | 152 | 134 | 3 | Tethyan Ammonite stratigraphy. Partly overwritten by Monteil, 1993 (for conversions see Supplement 1) |
| Monteil, 1993 | France | 152 | 140 | 3 | Some sections were calibrated to Tehyan ammonite stratigraphy, some only indicated stages (for conversions see Supplement 1) |
| Mudge and Bujak, 1996 | North Sea | 66 | 33 | 3 | Synthesis, using P planktonic foraminifer and NP nannoplankton events in the same section, but no data shown |
| Mudge and Bujak, 2001 | Faroe-Shetland | 66 | 54 | 3 | NP nannoplankton zones and P planktonic foraminifer zones in the same sections, but no data shown |
| Mudie, 1987 | North Atlantic | 8 | 0 | 1 | Magnetostratigraphy, N foraminifer strat and NN nannoplankton stratigraphy |
| Nikitenko et al., 2008 | Siberia | 150 | 134 | 3 | Siberian ammonite stratigraphy, in the paper correlated to Tethyan ammonite zones (for conversions see Supplement 1) |
| Nøhr-Hansen et al., 2002 | West Greenland | 66 | 62 | 3 | NP nannofossil stratigraphy on the same section |
| Nøhr-Hansen et al., 2020 | Greenland | 150 | 66 | 4 | Ammonite zonation on the same sections, but ammonite data shown separately. Calibrated to stages herein |
| Olde et al, 2015 | North Sea | 94 | 88 | 3 | Boreal ammonite stratigraphy on the same section |
| Oosting et al., 2006 | Australia | 131 | 120 | 4 | Tethyan ammonite stratigraphy on Angles, then inferred for Site 263 (for conversions see Supplement 1) |
| Pearce, 2010 | UK | 95 | 70 | 4 | UK ammonite zonations in nearby outcrops. Some intervals could not be correlated to the GTS2012 |
| Piasecki et al., 1992 | Greenland | 65 | 57 | 3 | NP Nannofossil stratigraphy on the same section |
| Poulsen and Riding, 2003 | North Sea, UK | 210 | 137 | 4 | Synthesis of Danish and British dinocyst events. Calibrated to Boreal ammonite stratigraphy, but presented, and herein plotted against stages |
| Poulsen, 1992 | Denmark | 163 | 145 | 4 | Boreal ammonite stratigraphy. Synthesis |



| | | | | | |
|---|---|---|---|---|---|
| Poulsen, 1998 | Poland | 169 | 164 | 3 | Boreal and Tethyan ammonite zones |
| Powell et al., 1996 | North Sea, UK | 59 | 55 | 1 | Magnetostratigraphy on the same sections |
| Powell, 1986 | NW Italy | 25 | 21 | 3 | NP nannofossil stratigraphy on the same section |
| Powell, 1988 | Central North Sea | 63 | 54 | 3 | NP nannofossil stratigraphy on the same sediments, no nannoplankton data directly shown |
| Powell in Powell, 1992 | North Sea, UK | 66 | 23 | 4 | P planktonic foraminifer and NP nannofossil stratigraphy, no direct calibration data shown |
| Prince et al., 2008 | UK | 89 | 83 | 3 | UK ammonite stratigraphy on the same sections, herein correlated to GTS2012 |
| Pross et al., 2010 | Italy | 35 | 22 | 1 | Magnetostratigraphy and independent age control from NP nannoplankton stratigraphy |
| Quaijtaal and Brinkhuis, 2012 | Alabama, USA | 37 | 30 | 1 | Magnetostratigraphy from the same section, independently established using nannoplankton and foraminifer stratigraphy |
| Quaijtaal et al., 2014 | Porcupine basin, Ireland | 17 | 11 | 1 | Magnetostratigraphy from the same section, independently established using nannoplankton stratigraphy |
| Radmacher et al., 2014a | Barentz Sea | 101 | 71 | 4 | Ages of the lithostratigraphic framework |
| Radmacher et al., 2014b | Zumaia, Spain | 74 | 70 | 1 | Magnetostratigraphy and UC nannoplankton stratigraphy on the same section |
| Radmacher et al., 2015 | Norwegian Sea | 113 | 66 | 4 | Regional lithostratigraphy dated using foraminifers and nannoplankton, but no direct independent age constraints shown |
| Riding and Helby, 2001a-g | NW Australia | 182 | 100 | 4 | Nannofossil and ammonite stratigraphy, but with some correlation to European and Tethyan sections (for conversions see Supplement 1) |
| Riding and Thomas, 1988 | UK | 160 | 150 | 3 | Boreal ammonite zonation on the same section (for conversions see Supplement 1) |
| Riding and Thomas in Powell, 1992 | North Sea | 202 | 145 | 4 | Boreal ammonite zonations, but not directly shown in paper (for conversions see Supplement 1) |
| Riding and Thomas, 1997 | N Scotland, isle of Skye | 166 | 155 | 3 | Boreal ammonite stratigraphy on the same section (for conversions see Supplement 1) |





| | | | | | |
|---|---|---|---|---|---|
| Riding et al, 2010 | Australia | 237 | 145 | 4 | Ammonites, conodonts, Belemnite/bivalve, NJ nannoplankton stratigraphy and strontium isotopes, but these data are not shown in the paper |
| Riley and Fenton, 1982 | UK/France | 166 | 160 | 3 | Boreal ammonite stratigraphy on the same sections |
| Schiøler, 1993 | Denmark | 72 | 66 | 4 | Stages, independent age constraints come from calcareous microplankton, not shown |
| Schreck and Matthiessen, 2014 | Northern Iceland | 14 | 5 | 1 | Magnetostratigraphy with NN nannoplankton and diatom stratigraphy |
| Schreck et al., 2012 | Northern Iceland | 15 | 2 | 1 | Magnetostratigraphy with NN nannoplankton and diatom stratigraphy |
| Schreck et al., 2013 | Northern Iceland | 15 | 2 | 1 | Magnetostratigraphy with NN nannoplankton and diatom stratigraphy |
| Schreck et al., 2017 | Northern Iceland | 15 | 2 | 1 | Magnetostratigraphy with NN nannoplankton and diatom stratigraphy |
| Shulgina et al., 1994 | Siberia | 145 | 132 | 3 | Boreal ammonite stratigraphy on the same sections (for conversions see Supplement 1) |
| Skupien, 2004 | Slovakia | 123 | 99 | 3 | Boreal ammonite stratigraphy (for conversions see Supplement 1) |
| Skupien and Vašíček, 2002 | Czech republic | 131 | 113 | 3 | Tethyan ammonite stratigraphy on the same section (for conversions see Supplement 1) |
| Slimani and Louwye, 2011 | Belgium | 75 | 62 | 4 | Regional lithostratigraphy calibrated to belemnite stratigraphy, tied to type Maastrichtian |
| Śliwińska et al., 2012 | Danish North Sea | 34 | 25 | 1 | Magnetostratigraphy and NP nannoplankton on the same section |
| Sluijs et al., 2003 | Tasmania | 37 | 30 | 2 | Magnetostratigraphy on the same section, but no independent chron assignment |
| Smelror et al., 1991 | Spain | 168 | 158 | 3 | Tethyan ammonite stratigraphy on the same section (for conversions see Supplement 1) |
| Smelror, 1988a | Greenland | 167 | 160 | 3 | Boreal ammonite stratigraphy on the same section (for conversions see Supplement 1) |
| Smelror, 1988b | Svalbard, Norway | 168 | 160 | 3 | NW European ammonite stratigraphy, herein calibrated to the Boreal zonation (for conversions see Supplement 1) |

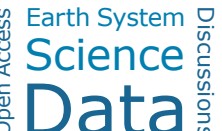

| Smelror, 1994 | Swabia | 167 | 165 | 4 | Ammonite and foraminifer stratigraphy, but herein calibrated against stages |
|---|---|---|---|---|---|
| Smelror and Dietl, 1994 | s Germany | 167 | 165 | 3 | Boreal ammonite stratigraphy on the same section |
| Smelror and Lominadze, 1989 | Caucasus | 166 | 163 | 3 | Boreal ammonite stratigraphy from the same section (for conversions see Supplement 1) |
| Soliman et al., 2012 | Golf of Suez, Egypt | 54 | 14 | 3 | NP/NN nannoplankton stratigraphy on the same section |
| Steeman et al., 2020 | Angola | 60 | 35 | 3 | P/E foraminifer stratigraphy on the same section |
| Stover and Hardenbol, 1994 | Belgium | 34 | 28 | 3 | NP nannoplankton stratigraphy on the same sections |
| Strauss and Lund, 1992 | Germany | 18 | 6 | 3 | Nannoplankton stratigraphy on the same sections |
| Thorn et al, 2009 | Seymour Island, Antarctica | 68 | 65 | 4 | The position of the K-Pg boundary |
| Tocher, 1987 | New Jersey Shelf, USA | 73 | 66 | 3 | Planktonic foraminifer stratigraphy on the same samples |
| Tocher and Jarvis, 1994 | Fumichon, France | 100 | 95 | 3 | Boreal ammonite stratigraphy on the same section |
| Tocher and Jarvis, 1995 | NW France | 101 | 92 | 3 | Boreal ammonite stratigraphy on the same section |
| Tocher and Jarvis, 1996 | NW France and SW UK | 110 | 95 | 3 | Boreal ammonite stratigraphy on the same section (for conversions see Supplement 1) |
| Torricelli, 2000 | southern Italy | 131 | 100 | 1 | Integrated bio-magneto-cyclostratigraphic framework, but only stages shown in the paper |
| Torricelli, 2006 | Piobbico, Italy | 113 | 100 | 3 | NC nannoplankton stratigraphy on the same section |
| Torricelli and Amore, 2003 | Southern Italy | 101 | 72 | 3 | CC nannoplankton stratigraphy on the same section |
| Torricelli et al., 2006 | Tremp Basin, Northern Spain | 53 | 51 | 3 | (P) planktonic foraminifer and NP nannoplankton stratigraphy on the same section |
| Türkecan et al., 2018 | Turkey | 18 | 14 | 3 | NN nannofossil and M foraminifer stratigraphy from the same section |
| Van de Schootbrugge et al., 2019a | UK, Arctic | 189 | 174 | 3 | Boreal ammonite stratigraphy on the same section |



| | | | | | |
|---|---|---|---|---|---|
| Van de Schootbrugge et al., 2019b | northern Germany | 202 | 178 | 3 | Boreal ammonite stratigraphy |
| Van Mourik and Brinkhuis, 2005 | Italy | 37 | 33 | 1 | Magnetostratigraphy on the same section |
| Van Mourik et al., 2001 | Offshore Florida | 39 | 35 | 1 | Magnetostratigraphy and CP nannoplankton stratigraphy on the same section |
| Van Simaeys et al., 2004 | Belgium | 33 | 22 | 3 | NP nannoplankton stratigraphy on the same sections |
| Van Simaeys et al., 2005 | Rhine Graben | 33 | 22 | 1 | Magnetostratigraphy on the same section |
| Vellekoop et al., 2015 | Tunisia | 67 | 65 | 3 | P foraminiferal zones on the same section |
| Versteegh, 1997 | North Atlantic Ocean, Italy | 3 | 2 | 1 | Isotope stages, herein recalibrated to NN and CN nannoplankton zones |
| Versteegh and Zevenboom, 1995 | South Italy | 28 | 0 | 1 | Magnetostratigraphy on the same section |
| Vieira et al., 2020 | North Sea | 59 | 56 | 3 | NP nannofossil and P foraminifer stratigraphy |
| Williams and Bujak, 1977 | Topical and North Atlantic Ocean | 25 | 0 | 5 | Stages, no independent age control given |
| Williams et al., 1993 | Northern Hemisphere | 210 | 0 | 5 | Stages, no independent age control given |
| Willumsen, 2012 | New Zealand | 70 | 64 | 3 | P foraminifer stratigraphy |
| Wilpshaar et al., 1996 | Central Italy | 35 | 22 | 3 | CP an NP nannoplankton and N planktonic foraminifer stratigraphy |
| Woollam and Riding, 1983 | UK | 209 | 140 | 3 | Boreal ammonite stratigraphy (for conversions see Supplement 1) |
| Wrenn and Kokinos, 1986 | Gulf Coast | 10 | 0 | 1 | Magnetostratigraphy on the same section |
| Zegarra and Helenes, 2011 | Equatorial Eastern Pacific | 18 | 0 | 1 | Independent age model from magnetostratigraphy, nannoplankton and foram stratigraphy |
| Zevenboom, 1995 | Italy | 26 | 16 | 3 | NP/NN nannoplankton stratigraphy on the same sections |





A decision tree is used to determine which papers to include into DINOSTRAT (Fig. 1). This tree first discards studies in which dinoflagellate cysts were the only stratigraphic tool to date the sequence. Although these papers do provide valuable information on stratigraphic order of events, discarding them from this review eliminates the risk of circular reasoning and inherited chronostratigraphic tie. Only those dinocyst events are included that could be calibrated against a stratigraphic tool that can be traced back to the bio-, magneto- or chronozones in the Geologic Time Scale 2012 (GTS2012; Gradstein et al., 2012). The decision tree distinguishes five tiers in these papers (Fig. 1):

- Tier 1 studies present dinocyst events along with magnetostratigraphic constraints obtained from the same sedimentary section. The interpretation of magnetochrons from the paleomagnetic signal was done without the use of dinoflagellate cyst biostratigraphy. Since magnetic reversals are globally synchronous, evaluating the synchroneity of dinocyst events with use of paleomagnetostratigraphy is most robust.

- Tier 2 studies present dinocyst events calibrated along with compromised or problematic magnetostratigraphic constraints on the same sedimentary section, for instance when the inclination signal suffers from a strong overprint, or when the magnetochron assignment is not clear. Studies in which dinocyst events served as biostratigraphic tool for magnetochron assignment are included in this tier as well.

- Tier 3 studies report dinocyst events together with biostratigraphic zones (from nannoplankton, foraminifer or ammonite zones), identified on the same sequence. These studies provide clear report on the identification of these zones in the sequence.

- Tier 4 studies report dinocyst events with biostratigraphy, of which either the derivation is unclear, or the tie to the GTS (e.g., for outdated ammonite zonations), or when biostratigraphic data does not come from the same sequence, but e.g., is interpreted from nearby outcrops.

- Tier 5 studies report dinocyst events with independent chronostratigraphy, of which the derivation is unverifiable, or represents a regional synthesis.



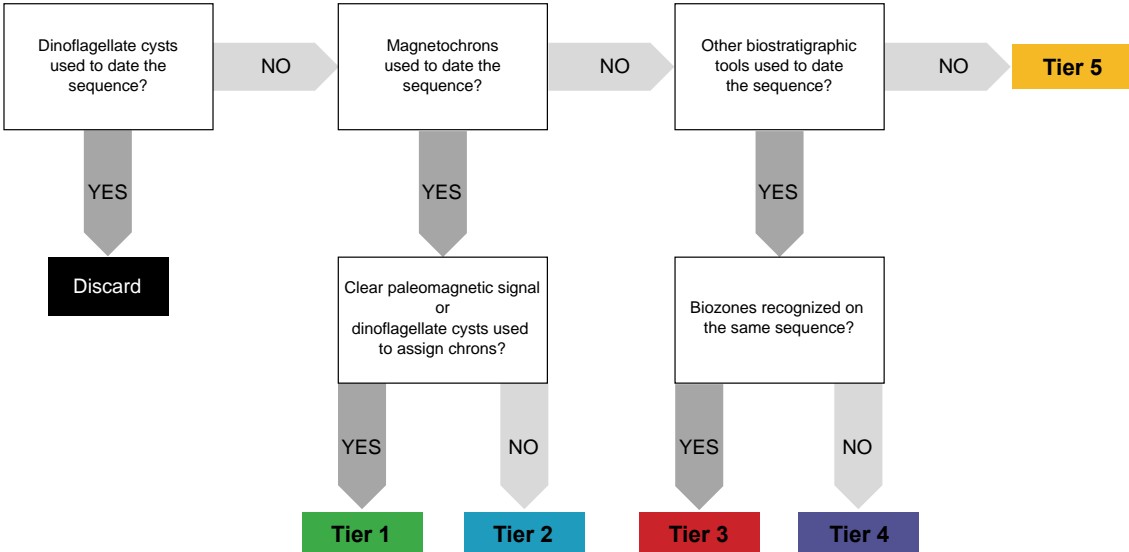

**Figure 1: Decision tree for including studies in this review, and categorization criteria for the 5 tiers.**

The absolute age of each dinocyst event is not explicitly entered into DINOSTRAT. Rather, its position within the zone it was calibrated to is entered. Ages are subsequently calculated via linear interpolation between these tie points, as follows:

135       [FO/LO] of [Genus, Species] = [##]% in ([stratigraphic tool]$[zone])       (1)

in which [##]% is linearly interpolated between base (0%) and top (100%) of tie points, [stratigraphic tool] is the bio-, magneto or chronozonation in the GTS2012, and [zone] is the name of the zone, chron or stage in which the dinocyst event falls. The rationale behind this approach instead of simple entry of the age is that while the absolute ages of dinocyst events are dependent

on the evolving knowledge of the chronostratigraphic time scale, the stratigraphic position of the event relative to the tie points in the record is fixed. This approach makes it easier to update the ages of the dinocyst events when the ages of the chrono-, magneto- and biozones are updated in the future. If dinocyst events fall between two different stratigraphic ties, the event is noted as follows:

145       [FO/LO] of [Genus, Species] = [##]% between [##]% in [stratigraphic tool]$[zone] and [##]% in [stratigraphic tool]$[zone]       (2)

Outdated Jurassic and Cretaceous ammonite zonation schemes are converted to those presented in the GTS2012 (see Supplement 1; following Ogg and Hinnov, 2012a, b and citations therein). FOs in the bottom of sections, and LOs at the top



of sections are systematically omitted, unless they were specifically indicated to represent an FO or LO. Younger publications presenting calibrations of dinocyst species from the same section overwrite older publications. Modern dinoflagellate cyst species and their latitudes (from Marret et al., 2020, and Mertens et al., 2014 for *Dapsilidinium pastielsii*) are entered with an LO of 0 Ma (modernst.csv in Bijl, 2021 for surface sediment station locations, modernsp.csv in Bijl, 2021 for dinoflagellate cyst species at those stations).


Each event entry in DINOSTRAT (Dinoevents_Jan2021.csv in Bijl, 2021) includes the (paleo-) latitude of that event. This is interpolated using the age of the event and its location, which has a paleolatitude evolution through time (Paleolatitude.csv in Bijl, 2021; with use of Paleolatitude.org; Van Hinsbergen et al., 2015). Paleolatitudes of sites in mobile orogenic belts are interpolated using regional tectonic reconstructions, and as such are prone to additional latitudinal uncertainty.


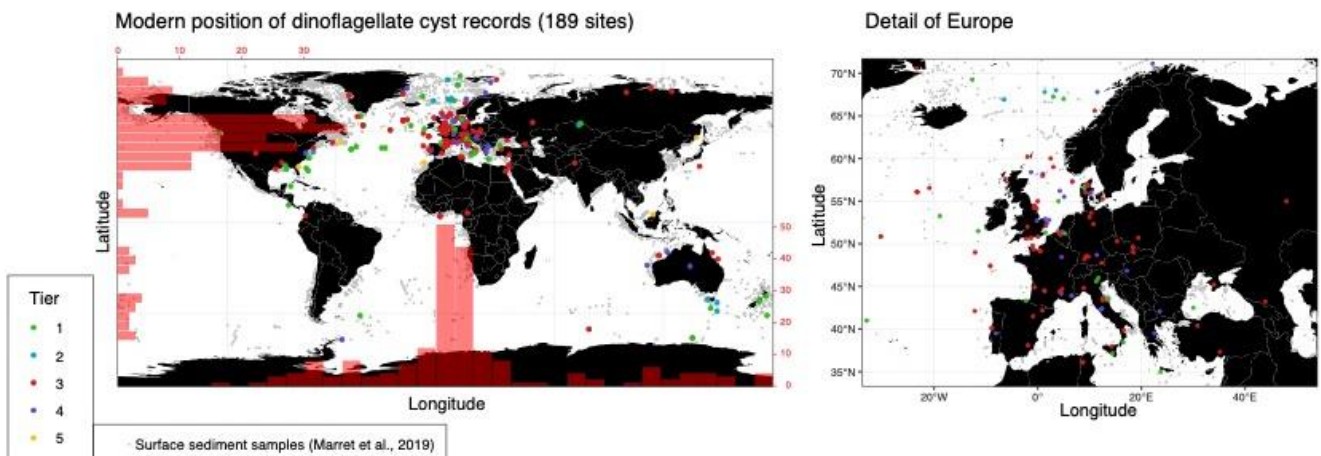

**Figure 2: Present-day geographic distribution of sedimentary sequences used in this study (colors of the dots correspond to the tier these sites belong in), and surface sediment stations (in grey dots; Marret et al., 2020 and Mertens et al., 2014). A. Global. B. Detailed map of sites in Europe.**

**3 The database**

**3.1 Sites**

DINOSTRAT currently contains dinocyst events from 199 publications and 189 sites. The wider North Atlantic/European area is strongly overrepresented (Fig. 2). Few sites are from the Pacific Ocean, southern Atlantic and Indian Ocean, and the equatorial region. Probably this reflects a genuine bias in the available information, because of focus of the community towards



economically interesting regions (e.g., for hydrocarbon industry). It may also in part reflect a bias towards 'western society' research, and poor accessibility of publications from non-western societies.

The paleolatitudinal position of the sites through time confirms the strong overrepresentation of Northern Hemisphere mid-latitude sections (Fig. 3), and underrepresentation of the tropical regions, Pacific Ocean and southern mid-latitudes. The Paleogene has the largest latitudinal spread of records, better yet than the Neogene. Particularly the Mesozoic has few entries

from the Southern Hemisphere and equatorial regions. The Mesozoic records are predominantly calibrated to ammonite stratigraphy (Tier 3 and 4), and in some occasions to magnetostratigraphy (Tier 1 and 2; Fig. 3). Ammonite zones presented in the papers often had to be converted to those in the GTS2012, which is not always straightforward, as the zone definitions have changed through time (Ogg and Hinnov, 2012a, b). The ammonite zonations are prone to regional diachroniety themselves, which was demonstrated particularly for the late Jurassic (Ogg and Hinnov, 2012b). This may create a level of

circular reasoning when dinocyst events are calibrated against these zones, because diachronous dinocyst events in DINOSTRAT may be the result of diachronous ammonite zones rather than diachronous dinocyst events.

### 3.2 Calibrated dinocyst events

DINOSTRAT version 1.0 includes over 8500 entries of calibrated dinoflagellate cyst events (excluding the modern dinocyst database). On a species level, originations in DINOSTRAT peak in the Middle Jurassic (Bajocian–Callovian) the lower

Cretaceous (upper Valanginian–Barremian) and the Eocene (Ypresian; Fig. 3b). Extinctions peak in the lower Cretaceous (Berriasian–Barremian), upper Cretaceous (Maastrichtian), Oligocene (Rupelian) and Miocene (Serravalian; Fig. 3b). This pattern is generally followed on a genus level, which likely has a stronger relation to the biologic diversity than dinocyst species diversity (Fensome et al., 1993).

The interpolated paleolatitudes for dinoflagellate cyst events in DINOSTRAT allows detailed evaluation of the latitudinal

synchroneity of dinocyst events. This paper presents a selection of the data in DINOSTRAT, focusing on the stratigraphic and geographic range of modern dinocyst species, of dinocyst families/subfamilies and of a selection of quasi-synchronous dinocyst events. Users can filter DINOSTRAT per locality (to present the stratigraphic order of events per site) and/or per taxon (to see the geographic variability of the range of any taxon), to serve their purposes.


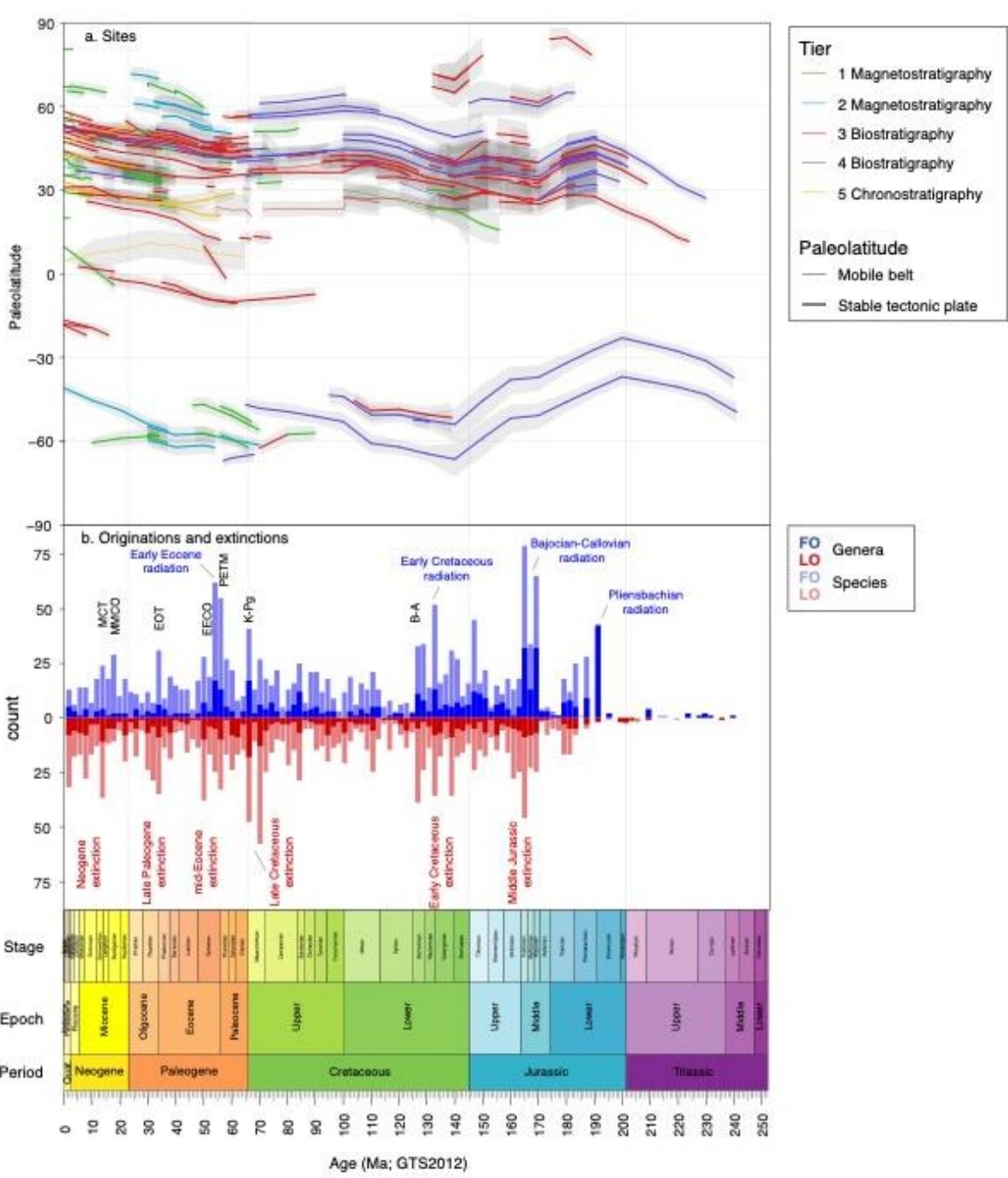

**Figure 3: Data in DINOSTRAT. a. Paleolatitude and age span of sites used in DINOSTRAT. Colors corresponds to tier, line thickness separates sites on stable oceanic or continental plates from those in mobile orogenic belts. Grey**





envelopes represent the error of the paleolatitude reconstruction inherited from the paleomagnetic reference frame (see Van Hinsbergen et al., 2015). b. Dinocyst events in DINOSTRAT, filtered for oldest FOs (blue) and youngest LOs (red) of dinocyst species (lighter shade) and genera (darker shade), in 2 Myr bins. Several phases of climate-environmental change are highlighted in black: Barremain Aptian boundary (B-A), Cretaceous-Paleogene boundary (K-Pg), Paleocene-Eocene Boundary (PETM), Early Eocene Climatic Optimum (EECO), Eocene-Oligocene transition (EOT), mid-Miocene Climatic Optimum (MMCO), Miocene climatic transition (MCT). Extinction and radiation phases in dinocysts are highlighted in red and blue text, respectively.

### 3.2.1 The stratigraphic range of modern dinoflagellate cyst species

Modern dinoflagellate cysts from surface sediment samples (Marret et al., 2020, n=3600 and Mertens et al., 2014, n=5) have a species-specific latitudinal spread. Sea surface temperature and nutrient conditions are the main controlling factors on modern assemblage compositions (Zonneveld et al., 2013). The database presented here allows comparison of modern latitudinal spread of these species to that of the past, and their age and latitude of oldest first occurrence (Supplement 2, and a selection in Fig. 4). Most modern species that have entries in DINOSTRAT have originations in the mid-Cenozoic: *Impagidinium* species, *Operculodinium centrocarpum*, *Tectatodinium pellitum*, *Tuberculodinium vancampoae* (Fig. 4). *Lingulodinium machaerophorum* has a first occurrence around 60 Ma. The exception is *Spiniferites ramosus*, a generalist species with a robust morphology through time, that has a remarkably consistent FO in the Berriasian (~145 Ma; Fig. 4). The dinocyst species that have geographic distributions restricted to one hemisphere today were also latitudinally restricted in the geologic past (e.g., *Spiniferites elongatus*, *Trinovantedinium variabile*; Fig. 4). *Achomosphaera andalousiensis, Dapsilidinium pastielsii, Impagidinium velorum, Melitasphaeridium choanophorum, Tectatodinium pellitum, Tuberculodinium vancampoae* had wider latitudinal distributions until the recent past, on both hemispheres. *Melitasphaeridium choanophorum* had progressively older LOs north and south of its restricted modern latitudinal distribution in northern mid-latitudes. *Lingulodinium machaerophorum* and *Polysphaeridium zoharyi* had a higher paleolatitudinal occurrence on only one hemisphere. Several modern taxa (e.g., *Bitectatodinium spongium*, *Polykrikos* spp., *Protoperidinium* spp., *Echinidinium* spp., most *Islandinium* species, most *Stelladinium* species, *Polarella glacialis*) have no entry yet in DINOSTRAT. This could be because some species concepts are relatively novel, or have poor preservation potential in the fossil record (e.g., because of selective degradation; Zonneveld et al., 2010).






**Figure 4: Age and paleolatitude of first (in blue) and last (in red) occurrences of selected modern dinoflagellate cyst species. Last occurrences come from both the surface sediment database (Marret et al., 2020, with Mertens et al., 2014) and entries in DINOSTRAT.**

### 3.2.2 Dinocyst (sub-) families

Range charts of the Sites in DINOSTRAT are provided in the Supplements (see "Sites" folder in Supplement 2). The age over paleolatitude entry in DINOSTRAT allows evaluation of the latitudinal difference in event ages for each individual species in DINOSTRAT (n=1914), as well as for groupings per genus (n=460) and family (n=28) (Supplement 2). Users can produce/adapt these plots themselves with help of the R markdown script "plot creator.Rmd" in Bijl, 2021). The most robust dinocyst events will have synchronous ages of FOs and LOs per paleolatitude (i.e., vertical blue and red lines in the plots of

Supplement 2). The FOs and LOs connected per species and grouped in (sub)families are plotted and described below, with particularly synchronous taxa highlighted. The purpose of these plots is threefold: First, they show the total stratigraphic range and latitudinal spread of these dinoflagellate (sub)families, and time intervals when and where phases of strong diversification and extinction occur in that (sub)family. Second, as with the plots of modern species, they show in which paleolatitudes these supra-generic groups first appear, but also where they last go extinct. Although earlier compilations of the evolution of dinocyst

families do exist (e.g., McRae et al., 1996), DINOSTRAT presents the fundamental spatio-temporal observations that underpin these compilations. Thirdly, the plots allow presentation of the database in a way that the validity of extrapolating dinoflagellate cyst events on a supra-regional scale can be critically evaluated in the discussion.

Order Gonyaulacales

Family Areoligeraceae (Fig. 5)

*Range:* The Areoligeraceae range from the Bathonian (~168 Ma, FOs of *Adnatosphaeridium* spp. and *Senoniasphaera* spp.) to the mid-Miocene (~18 Ma, LO of *Chiropteridium galea*). Areoligeraceae seem to range longer in Northern Hemisphere mid-latitudes (FO ~169 Ma; LO ~18 Ma) than in the rest of the world (FO ~145 Ma; LO ~36Ma), although this can be in part related to a sampling bias. The oldest FOs in NH mid-latitudes are species with a stratigraphic occurrence restricted to that

area.

*Quasi-synchronous events:* Events of species of *Areoligera, Chiropteridium, Glaphyrocysta, Palynodinium, Schematophora and Senoniasphaera,* particularly in the late Cretaceous and Paleogene (Fig. 5). Many taxa in this subfamily however show strongly diachronous events between hemispheres.



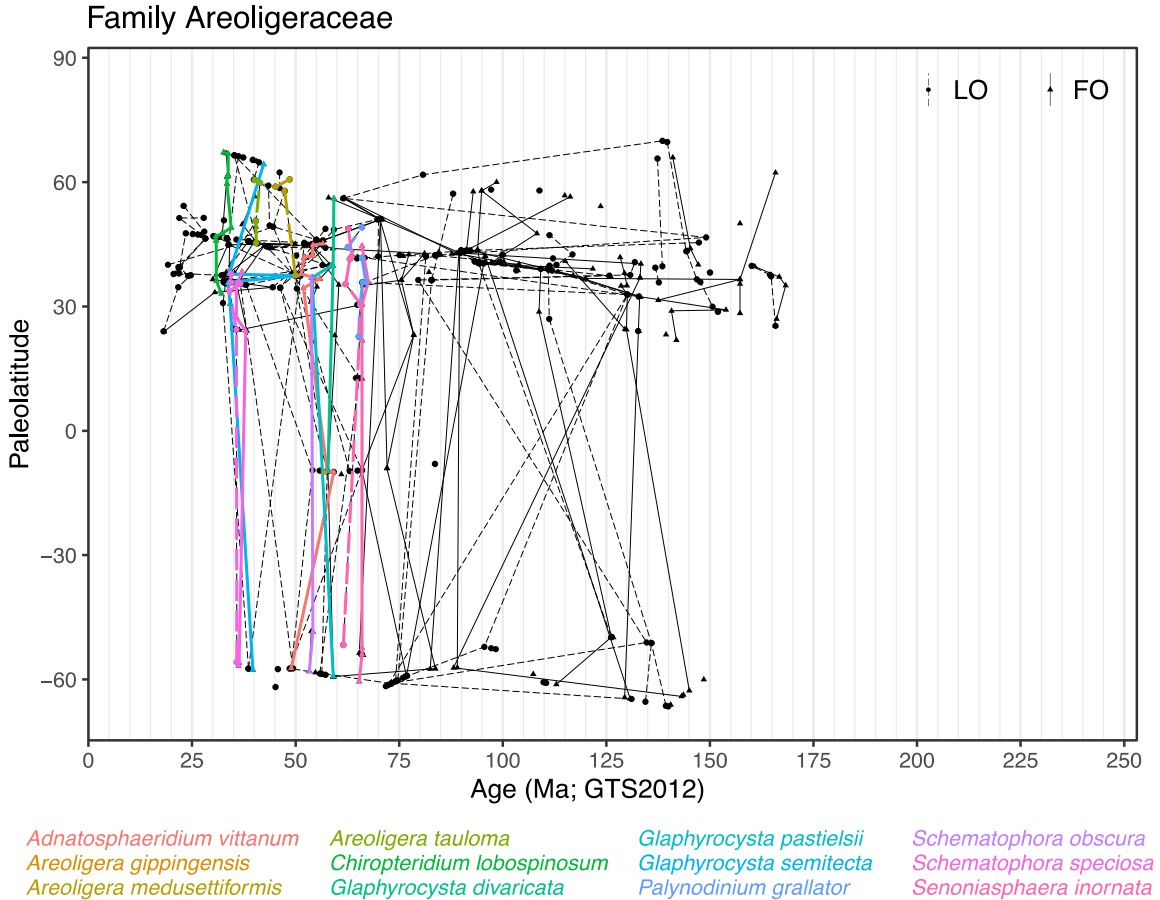

**Figure 5: Ages and paleolatitudes of first (solid line and triangles) and last (dashed line and circles) occurrences of dinocyst species of the Family Areoligeraceae. Solid and dashed lines connect first and last occurrences, respectively, for each species, between sites. Colored lines represent quasi-synchronous species events.**

Family Ceratiaceae (Fig. 6)

*Range:* The Ceratiaceae first appear in the Tithonian (~152Ma, FO of *Muderongia simplex*) in NH mid-latitudes, represents a diverse group in the early Cretaceous and has an LO in the latest Cretaceous (~66 Ma, LO of *Odontochitina operculata*).

*Quasi-synchronous events:* LO *Odontochitina costata,* LO *Phoberocysta neocomica,* range of *Pseudoceratium pelliferum* (Fig. 6).



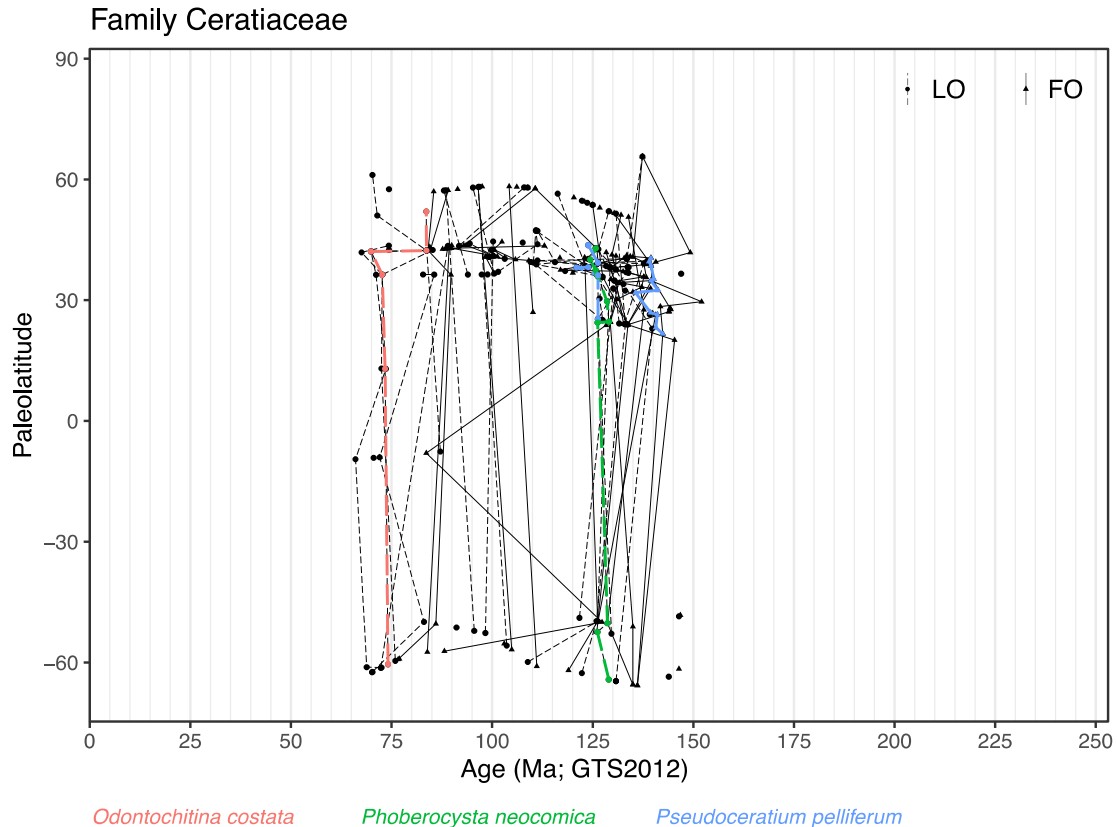

**Figure 6: As Figure 5, but for the Family Ceratiaceae.**

Family Cladopyxiaceae (Fig. 7)

*Range:* This family first appears in the Pliensbachian (~188Ma, FO of *Freboldinium* spp.), and ranges until the late Oligocene (~25 Ma; *Licracysta semicirculata*).

*Quasi-synchronous events:* Several species of *Enneadocysta*. LO of *Fibradinium annetorpense* around 60 Ma and the LO of *Licracysta semicirculata* around 26 Ma Most entries in the late Cretaceous and Paleogene are highly diachronous.

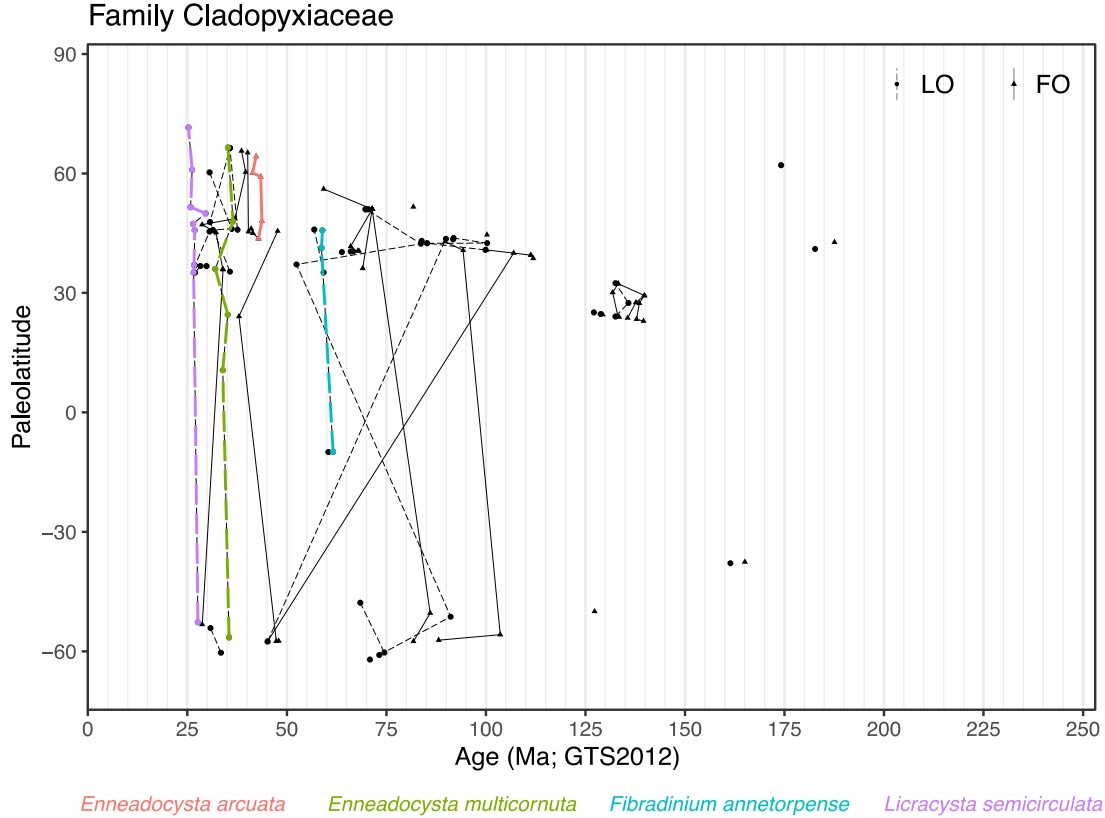

Figure 7: As Figure 5, but for the Family Cladopyxiaceae.

Family Goniodomaceae (Fig. 8)

*Range:* Goniodomaceae first appear in the mid-Tithonian (~150 Ma, FO of *Hystrichosphaeridium petilum*) in the NH mid-latitudes, most entries are from the Paleogene, and continue with modern species *Polysphaeridium zoharyi* and *Tuberculodinium vancampoae. Quasi-synchronous events:* Species of *Alisocysta, Eisenackia, Heteraulacacysta* and *Homotryblium.* Many species ranges in this family are notably diachronous. Although some species do seem to show similar event ages between southern high latitudes and northern mid-latitudes (Fig. 8), those with multiple entries in the northern mid-latitudes, where site density is highest, show strong diachroneity over short latitudinal distances. Modern species have a restricted latitudinal spread to subtropical and tropical regions, but not too long into the geologic past, species of this family exhibited much wider latitudinal ranges (65°S – 70°N).

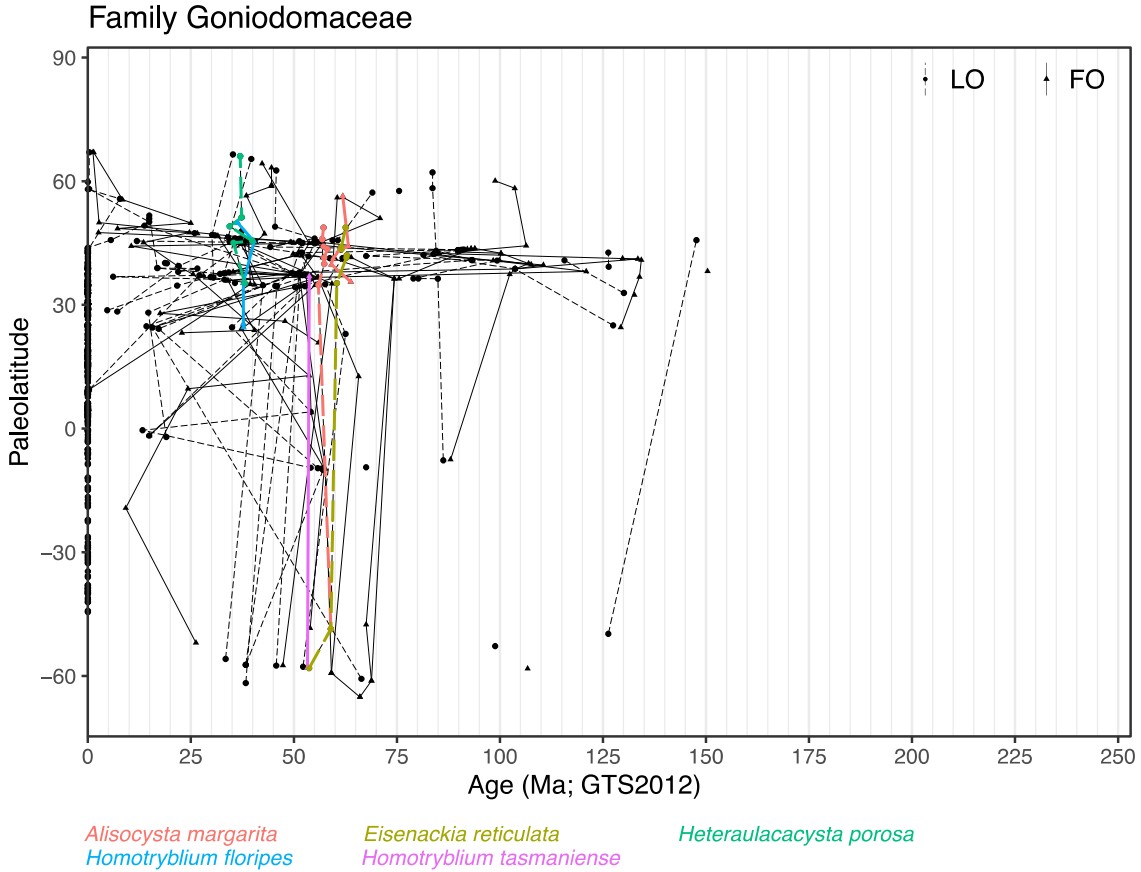

**Figure 8: As Figure 5, but for the Family Goniodomaceae.**


Family Gonyaulacaceae

Subfamily Cribroperidinioideae (Fig. 9)

*Range:* This subfamily includes the extant species *Operculodinium centrocarpum* and *Lingulodinium machaerophorum*. The subfamily first appears in NH mid-latitudes in the Aalenian (~172 Ma) with *Kallosphaeridium* spp. and in the Bajocian

(~169Ma) with *Cribroperidinium* spp., and shortly thereafter *Aldorfia* and *Korystocysta*. *Cribroperidinium* is a long-ranging genus. Many entries are from the early Cretaceous (~125 Ma) and early Paleogene (66–34 Ma)

*Quasi-synchronous events:* Several species of *Cordosphaeridium* and *Danea*, and species of *Aldorfia, Apteodinium, Carpatella, Cooksonidinium, Diphyes, Hystrichokolpoma* and *Operculodinium*. The subfamily has many entries in the Paleogene, but many of these events are not synchronous latitudinally.




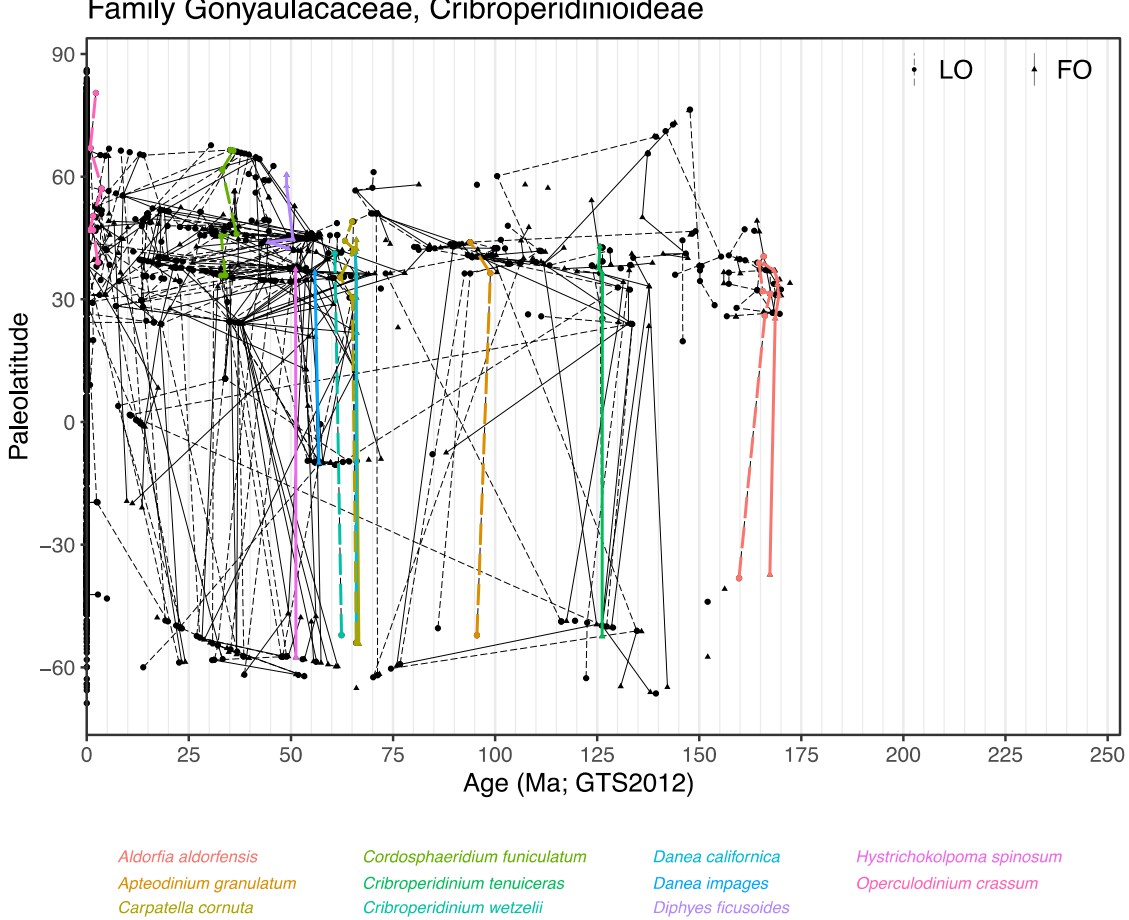

**Figure 9: As Figure 5, but for the Family Gonyaulacoideae, subfamily Cribroperidinioideae.**

Subfamily Gonyaulacoideae (Fig. 10)

*Range:* The subfamily of Gonyaulacoideae includes common modern cyst genera such as *Spiniferites* spp., *Achomosphaera* spp., *Impagidinium* spp., *Nematosphaeropsis* spp. and *Tectatodinium* spp. The subfamily first occurs in the Bajocian (~170 Ma), with the FO of *Gonyaulacysta* spp. and *Tubotuberella* spp.

*Quasi-synchronous events:* species of *Achomosphaera, Ataxiodinium, Callaiosphaeridium, Corrudinium, Ectosphaeropsis Hystrichodinium, Impagidinium, Spiniferites and Unipontidinium* (Fig. 10). Events of species of *Escharisphaeridia* spp.,

*Gonyaulacysta* spp., and *Tubotuberella* spp., range slightly longer in Northern Hemisphere high latitudes than in mid-latitudes. Many species in this subfamily are strongly diachronous.

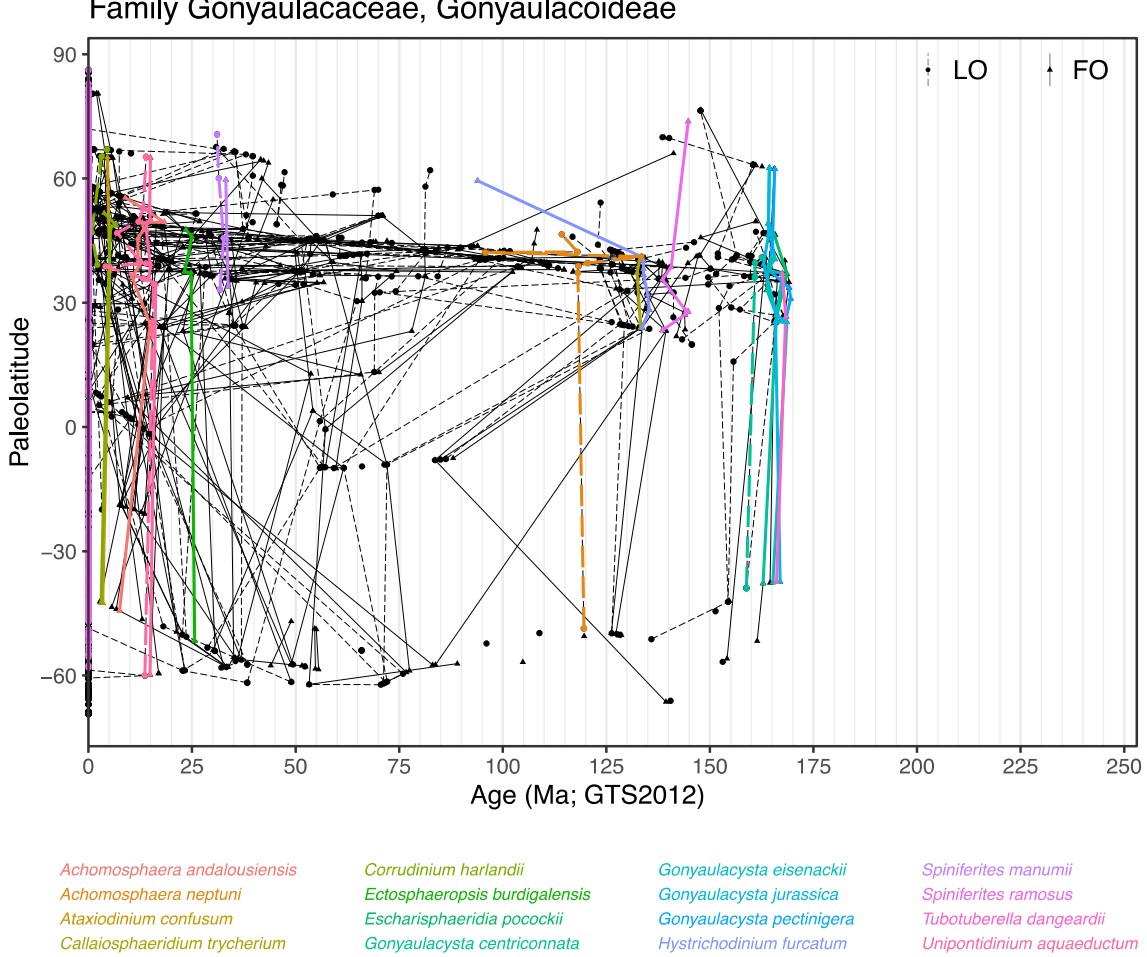

Figure 10: As Figure 5, but for the Family Gonyaulacaceae, subfamily Gonyaulacoideae.


Subfamily Leptodinioideae (Fig. 11)

*Range:* The Leptodinioideae first appear in the Aalenian (~172 Ma, FO of *Meiourogonyaulax valensii*), and includes many species events in the Bajocian and Bathonian. Although most entries are in the Jurassic and lower Cretaceous, the subfamily ranges into the late Miocene (~8 Ma, LO of *Acanthaulax miocenica*).

*Quasi-synchronous events:* Events in species of *Ambonosphaera*, *Areosphaeridium* (NH), *Cooksonidium, Ctenidodinium, Dichadogonyaulax, Endoscrinium, Herendeenia, Kleithriasphaeridium, Leptodinium*, *Limbodinium, Litosphaeridium*, *Rigaudella aemula*, Sirmiodiniopsis, *Stiphrosphaeridium* and *Wanaea*.

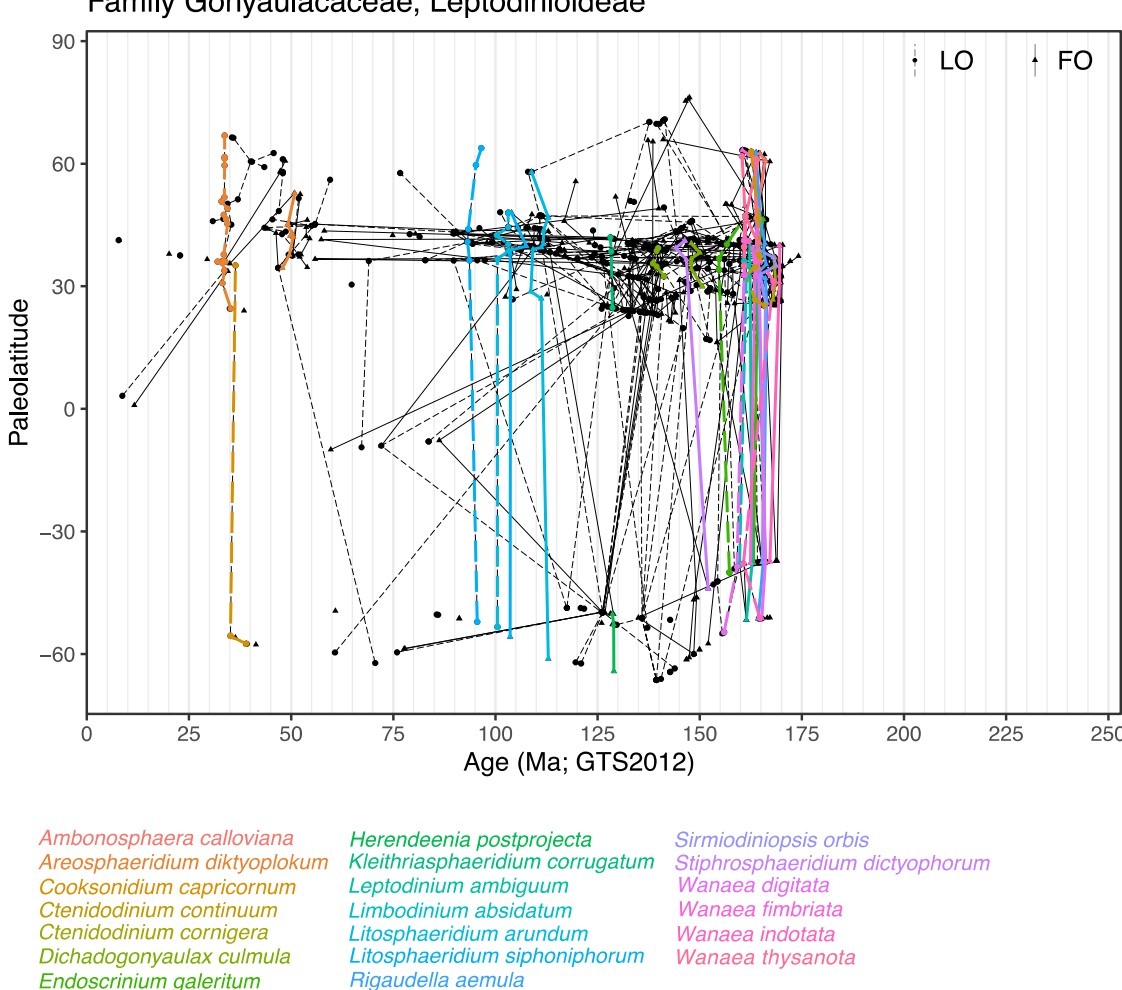

**Figure 11: As Figure 5, but for the Family Gonyaulacoideae, subfamily Leptodinioideae.**

Other (Fig. 12)

*Remarks:* Other species in the Family Gonyaulacaceae could not be assigned to a subfamily. Species of Barbatacysta, *Chytroeisphaeridia*, *Glossodinium*, Hemiplacophora, Nelchinopsis, Saturnodinium, *Scriniodinium*, *Sepispinula*, *Stephodinium, Trichodinium* spp. have remarkably consistent events.

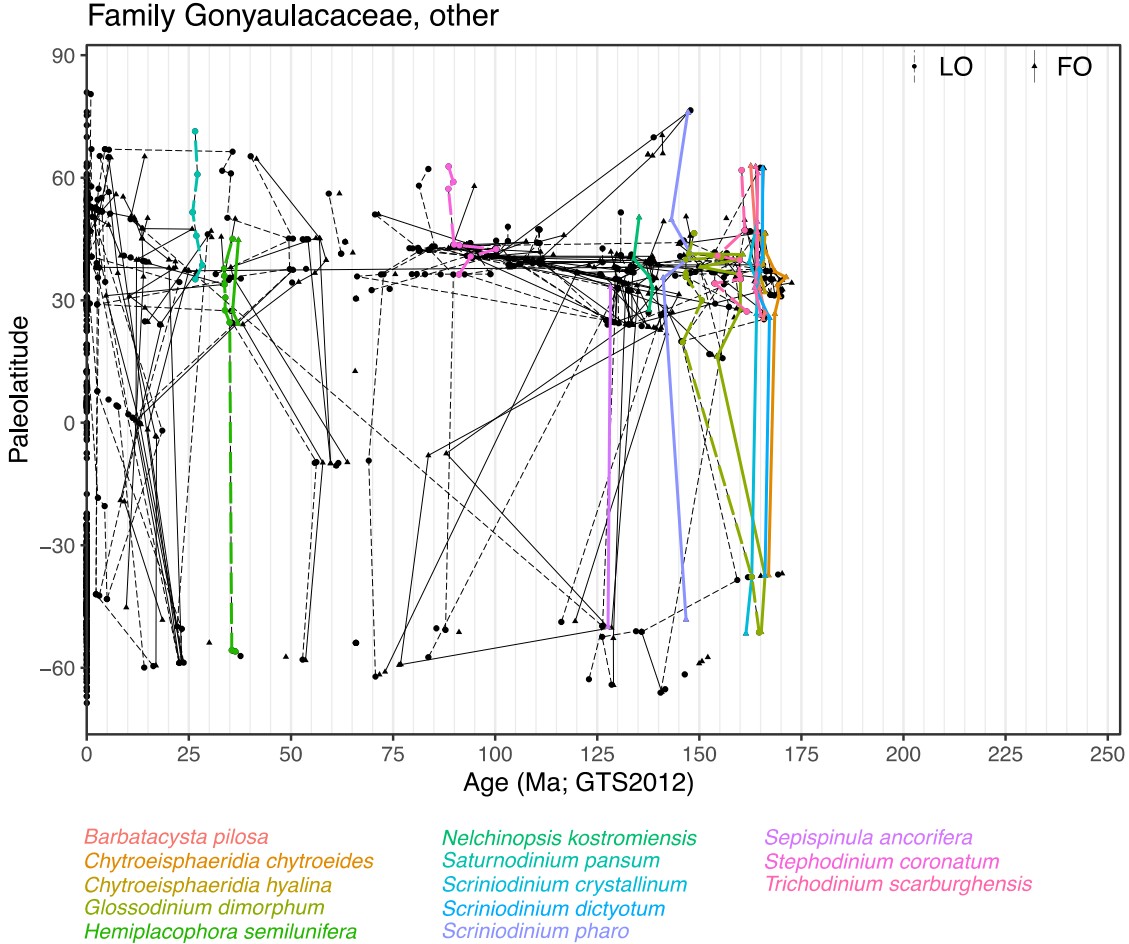

**Figure 12: As Figure 5, but for other genera in the Family Gonyaulacoideae.**

Family Mancodiniaceae (Fig. 13)

*Range:* Species of Mancodiniaceae occur in sediments from the late Sinemurian (~190 Ma) to the mid-Bathonian (~167 Ma) and seem quasi-synchronous latitudinally.



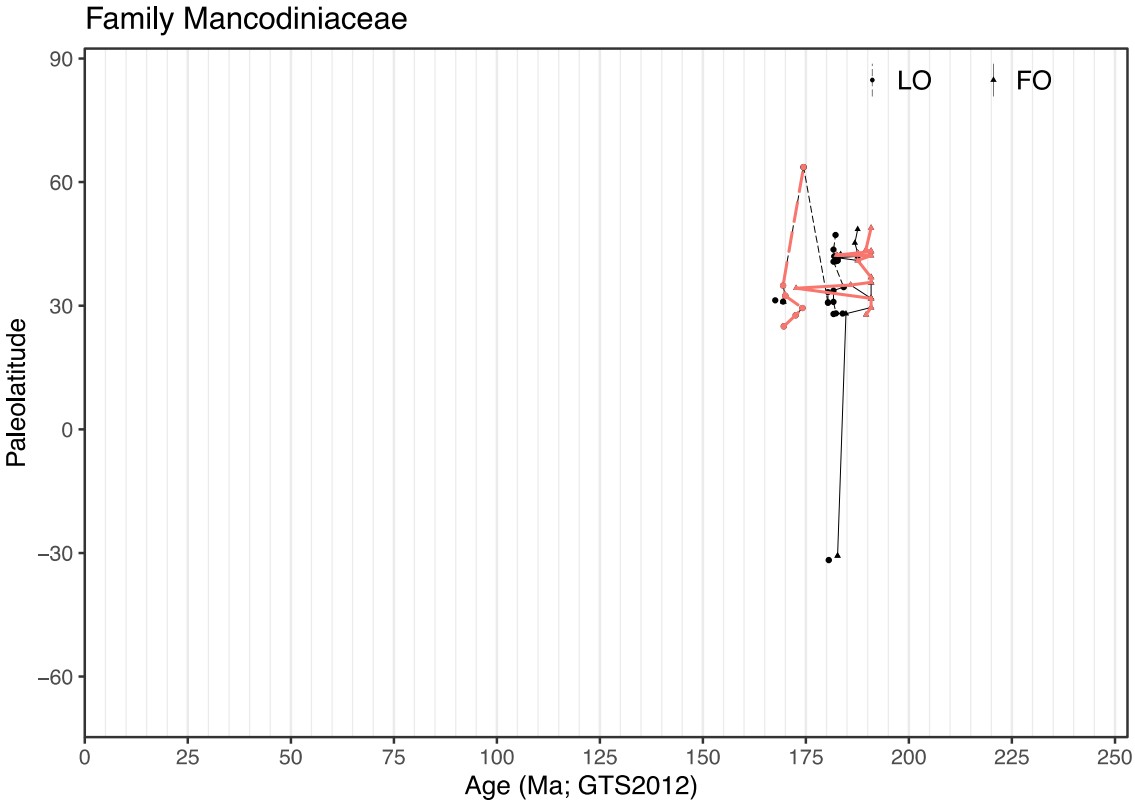

Figure 13: As Figure 5, but for the Family Mancodiniaceae.

Family Pareodiniaceae (Fig. 14)

*Range:* Pareodiniaceae first appear in the late Toarcian (~176 Ma, FO of *Pareodinia halosa*) and range in northern hemisphere mid-latitudes into the Cenomanian (~95 Ma, LO of *Batioladinium jaegeri*). Events of species in *Carpathodinium, Pareodinia* (both NH only), *Aprobolocysta* and *Batioladinium* appear quasi-synchronous.

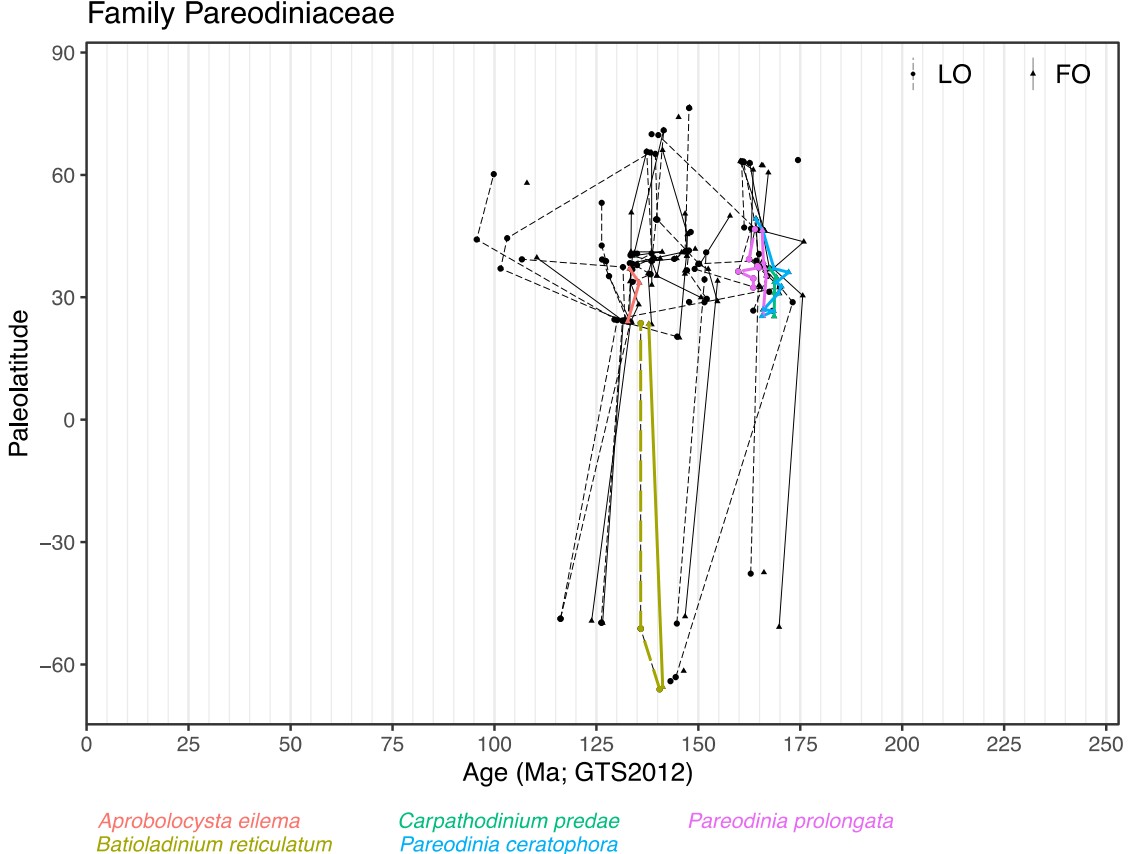

**Figure 14: As Figure 5, but for the Family Pareodiniaceae.**

Family Scriniocassiaceae (Fig. 15)

*Range:* Scriniocassiaceae range from the Pliensbachian (~187 Ma, FO of *Scriniocassis weberi*) to the Bajocian (~169Ma, LO of *Scriniocassis weberi*) and comprise of only 3 species. Events from this family are only reported from the Northern Hemisphere.

Earth System
Science
Data

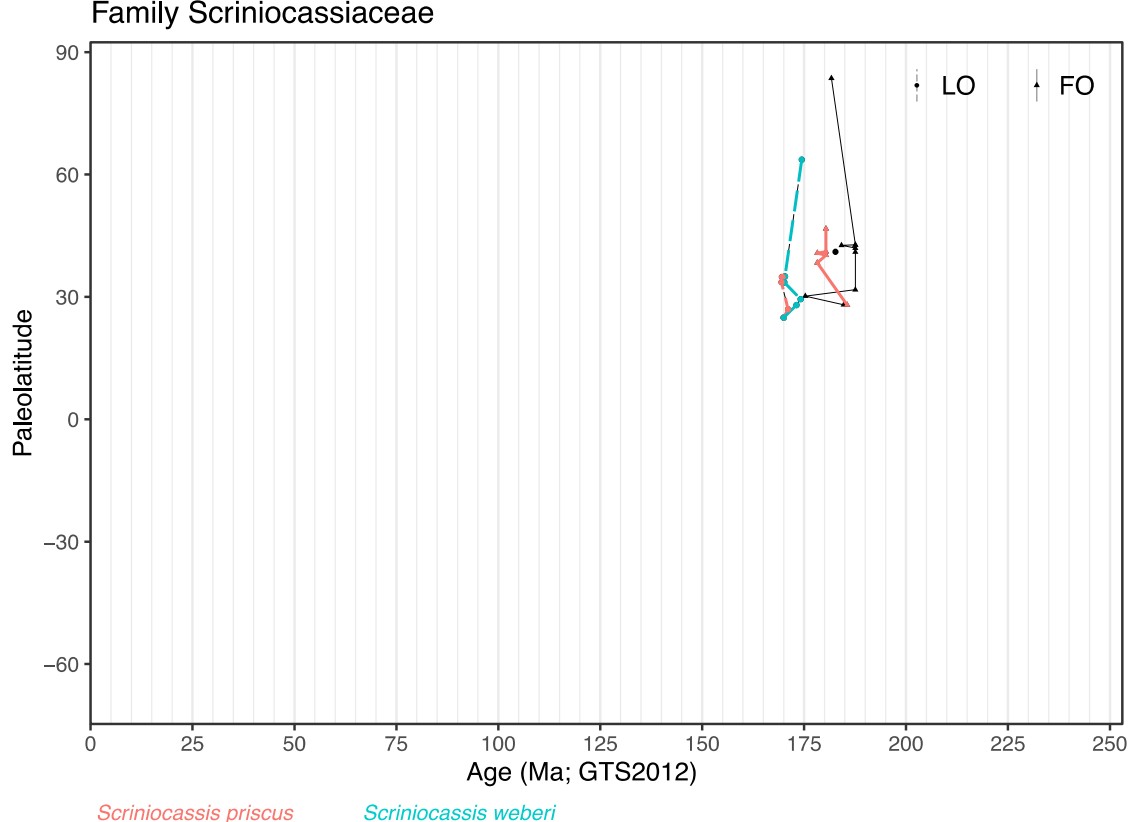

**Figure 15: As Figure 5, but for the Family Scriniocassiaceae.**

Family Shublikodiniaceae (Fig. 16)

*Range:* Cysts from the Family Shublikodiniaceae occur in the late Triassic (FO of *Rhaetogonyaulax wigginsii* in the Carnian,

~230 Ma) to early Jurassic (LOs of *Dapcodinium sacculus* and *Dapcodinium ovale* in the mid-Pliensbachian, 187 Ma).

*Quasi-synchronous events:* LO of *Rhaetogonyaulax rhaetica* close to the Triassic-Jurassic Boundary, LO of *Dapcodinium priscum.*

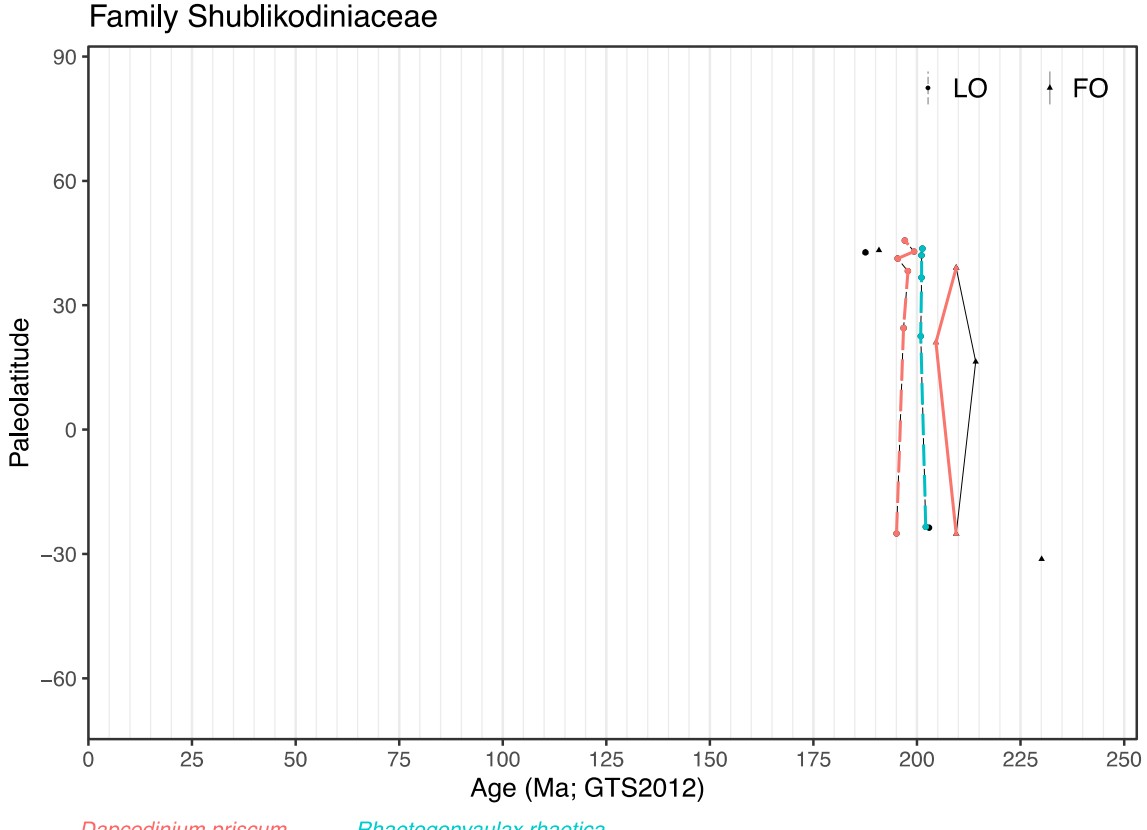

**Figure 16: As Figure 5, but for the Family Shublikodiniaceae.**

Family uncertain (Fig. 17)

*Remarks*: This group of which the family is uncertain does contain several stratigraphically synchronous species (Fig. 17). Ranges of species of Amiculosphaeridia, *Atopodinium, Batiacasphaera, Cleistosphaeridium, Dingodinium, Distatodinium, Heslertonia, Labyrinthodinium, Membranilarnacia, Mendicodinium, Oligokolpoma* and *Valensiella* are quasi-synchronous.



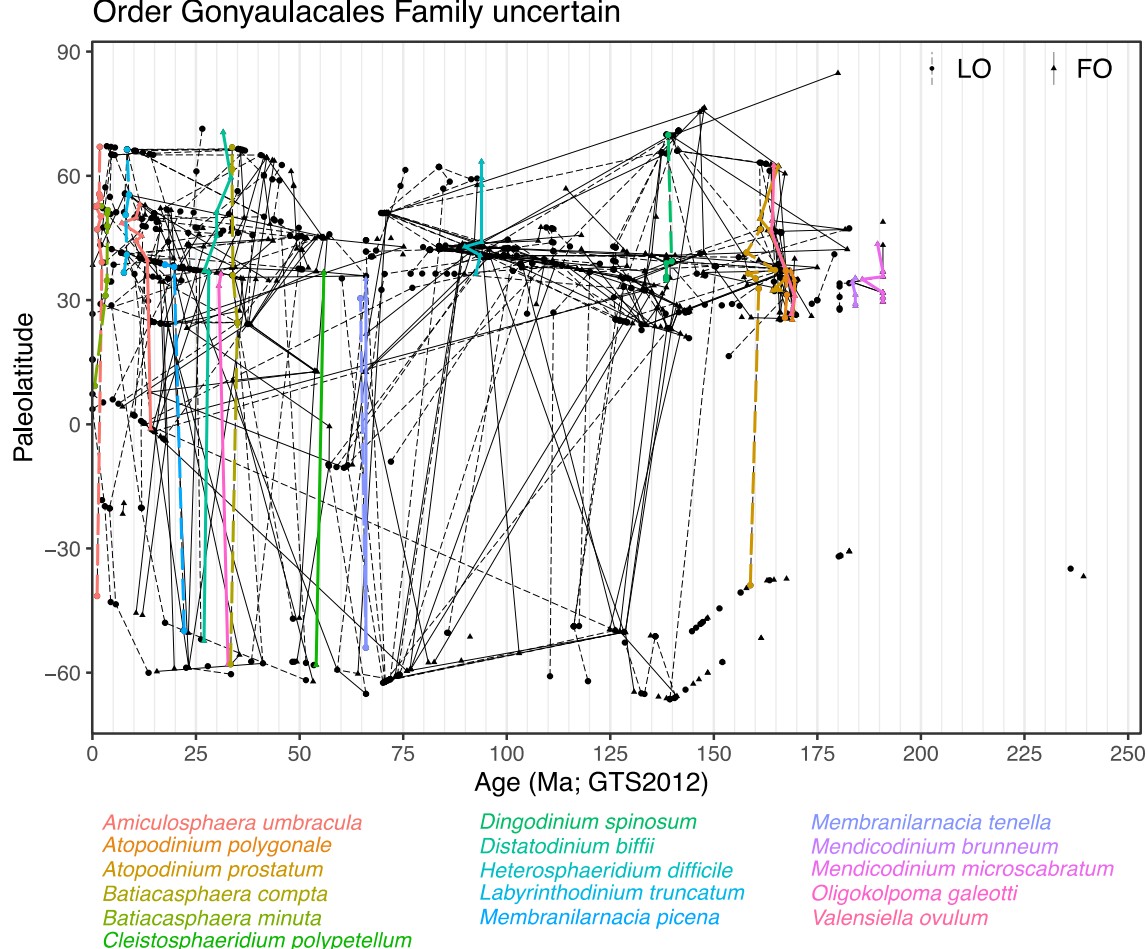

**Figure 17: As Figure 5, but for the Order Gonyaulacales, Family uncertain.**

Order uncertain

Family Comparodiniaceae (Fig. 18)

*Range:* Cysts from this Family range from the late Sinemurian (191 Ma, FO of *Valvaeodinium* spp.) to the mid-Valanginian (134 Ma, LO of *Biorbifera johnwingii*). All species except *Valvaeodinium spinosum* and *Biorbia ferox* have ranges restricted to the Northern Hemisphere.

*Quasi-synchronous events:* Range of *Biorbifera johnwingii*, FO of *Valvaeodinium spinosum*, LO of *Valvaeodinium koessenium*.

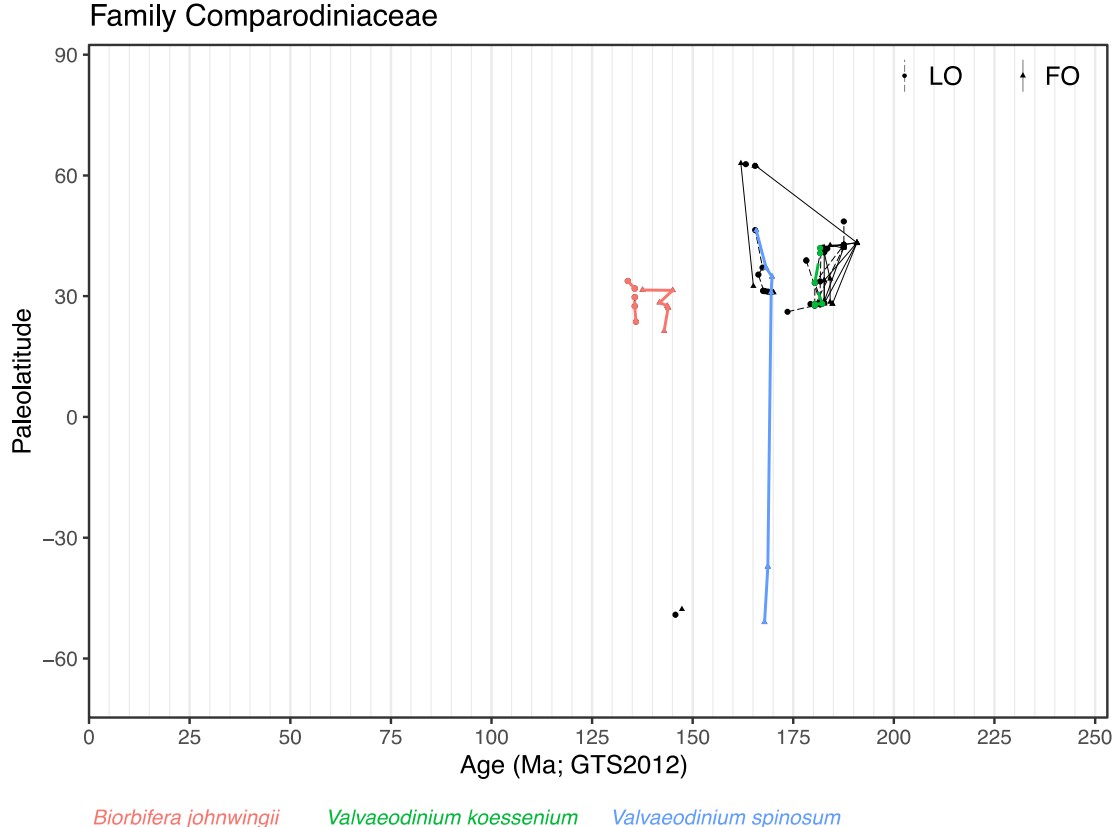

**Figure 18: As Figure 5, but for the Family Comparodiniaceae.**


Family Stephanelytraceae (Fig. 19)

*Range:* Stephanelytraceae cysts comprine of one genus, which ranges from the Callovian (~166 Ma) to the late Aptian (~117 Ma), and seems quasi-synchronous.

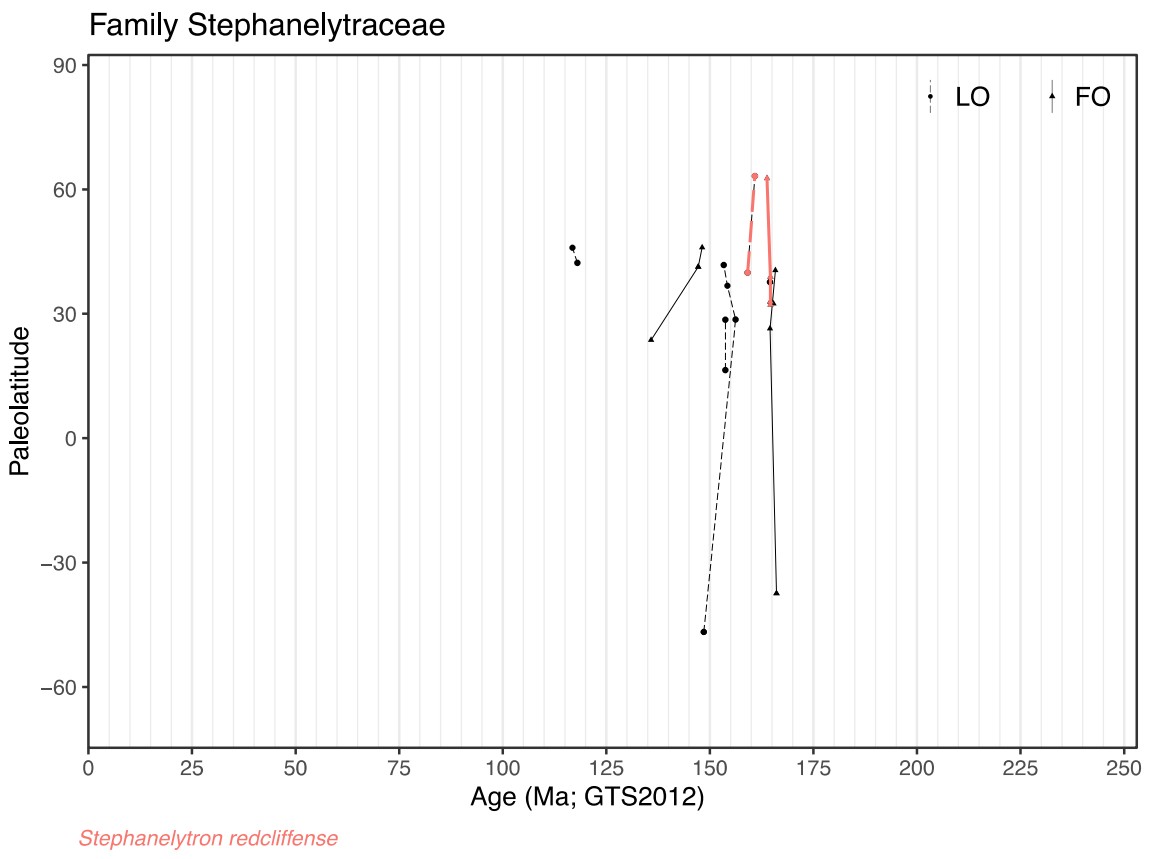

Stephanelytron redcliffense


**Figure 19: As Figure 5, but for the Family Stephanelytraceae.**

Order Peridiniales

Family Heterocapsaceae (Fig. 20)

*Range:* Heterocapsaceae range from the mid-Sinemurian (195 Ma, FO of *Liasidium variabile*) to the mid-Albian (105 Ma, LO of *Angustidinium acribes*).

*Quasi-synchronous events:* Range of *Liasidium variabile* and *Parvocysta bullala,* restricted to Northern Hemisphere mid-latitudes.

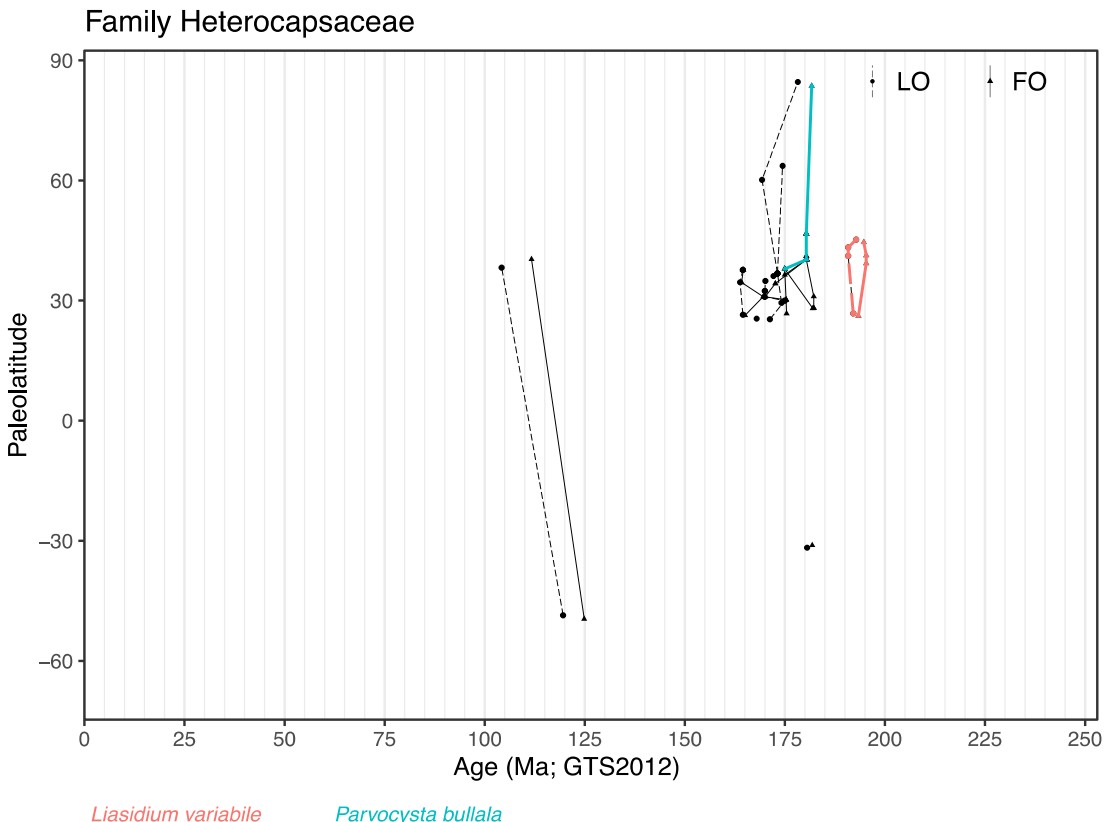

Figure 20: As Figure 5, but for the Family Heterocapsaceae.

Family Peridiniaceae

Subfamily Deflandreoideae (Fig. 21)

*Range:* Deflandroideae first occur on the Southern Hemisphere in the Oxfordian (~ 161 Ma) with *Pyxidiella* spp. *Isabelidinium* and *Eurydinium* first appear in the Albian (~109 Ma), and many species first appear in the late Cretaceous (~95–66 Ma). The subfamily goes extinct with the LO of *Sumatradinium* spp. around 5 Ma and appears to range longest in low and mid-latitudes. Deflandeoideae have many FO and LO entries on both hemispheres, particularly in the Late Cretaceous and early Paleogene.

*Quasi-synchronous events:* Several species of *Cerodinium*, *Manumiella, Trithyrodinium*, and *Isabelidinium* have synchronous

events in the Maastrichtian-Paleocene (70–60 Ma).

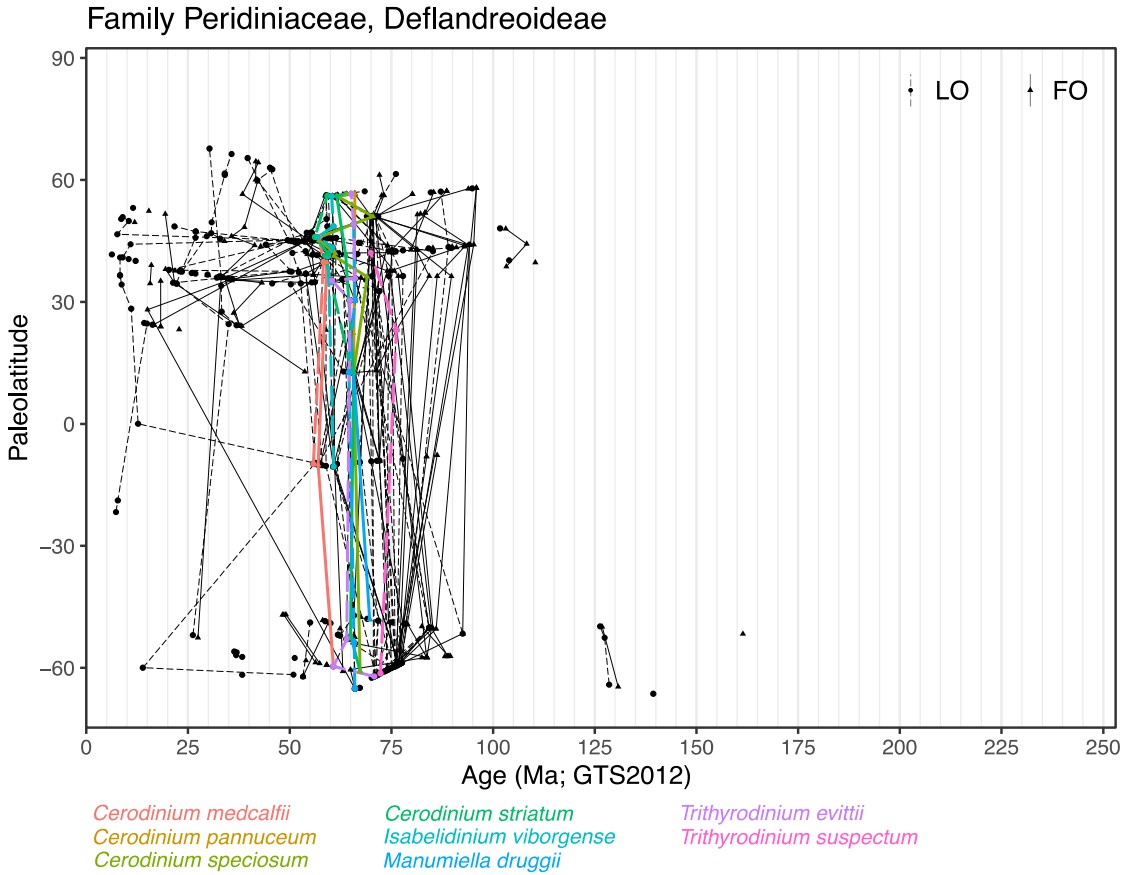

**Figure 21: As Figure 5, but for the Family Peridiniaceae, subfamily Deflandreoideae.**

Subfamily Palaeoperidinioideae (Fig. 22)

*Range:* The Palaeoperidinioideae range from the mid-Valanginian (~135 Ma, FO of *Subtilisphaera perlucida*) to the late

Oligocene (~26 Ma, LO of *Phthanoperidinium comatum*).

*Quasi-synchronous events:* Range of *Palaeoperidinium pyrophorum* and the LO of *Phthanoperidinium comatum*.

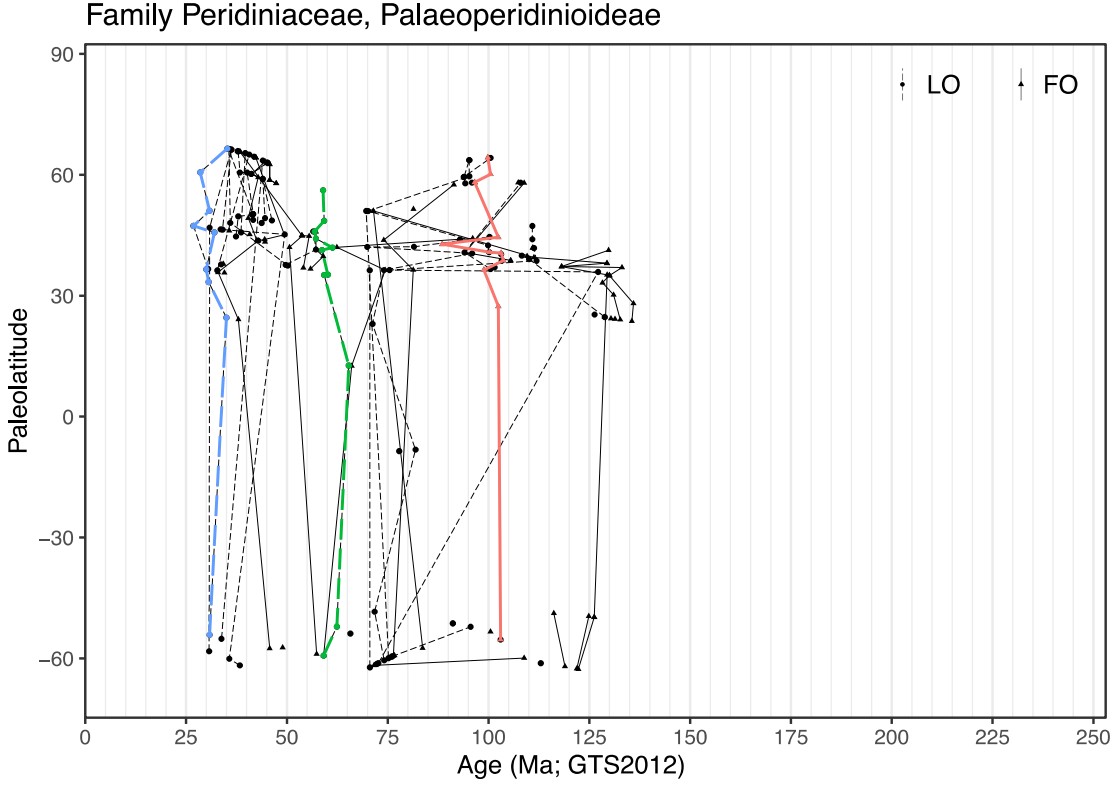

**Figure 22: As Figure 5, but for the Family Peridiniaceae, subfamily Palaeoperidinioideae.**

Subfamily Wetzelielloideae (Fig. 23)

*Range:* Wetzelielloideae range from the mid-Paleocene (~62 Ma, FO of *Apectodinium homomorphum*) to the late Oligocene (~23 Ma, LO of *Wetzeliella symmetrica*). Diversification particularly in the Ypresian leads to many species with short stratigraphic ranges, many of which are relatively synchronous latitudinally. Several species appear to range longer in the Northern Hemisphere than on equal paleolatitudes on the Southern Hemisphere. Many species lack chronostratigraphic tie in equatorial records.

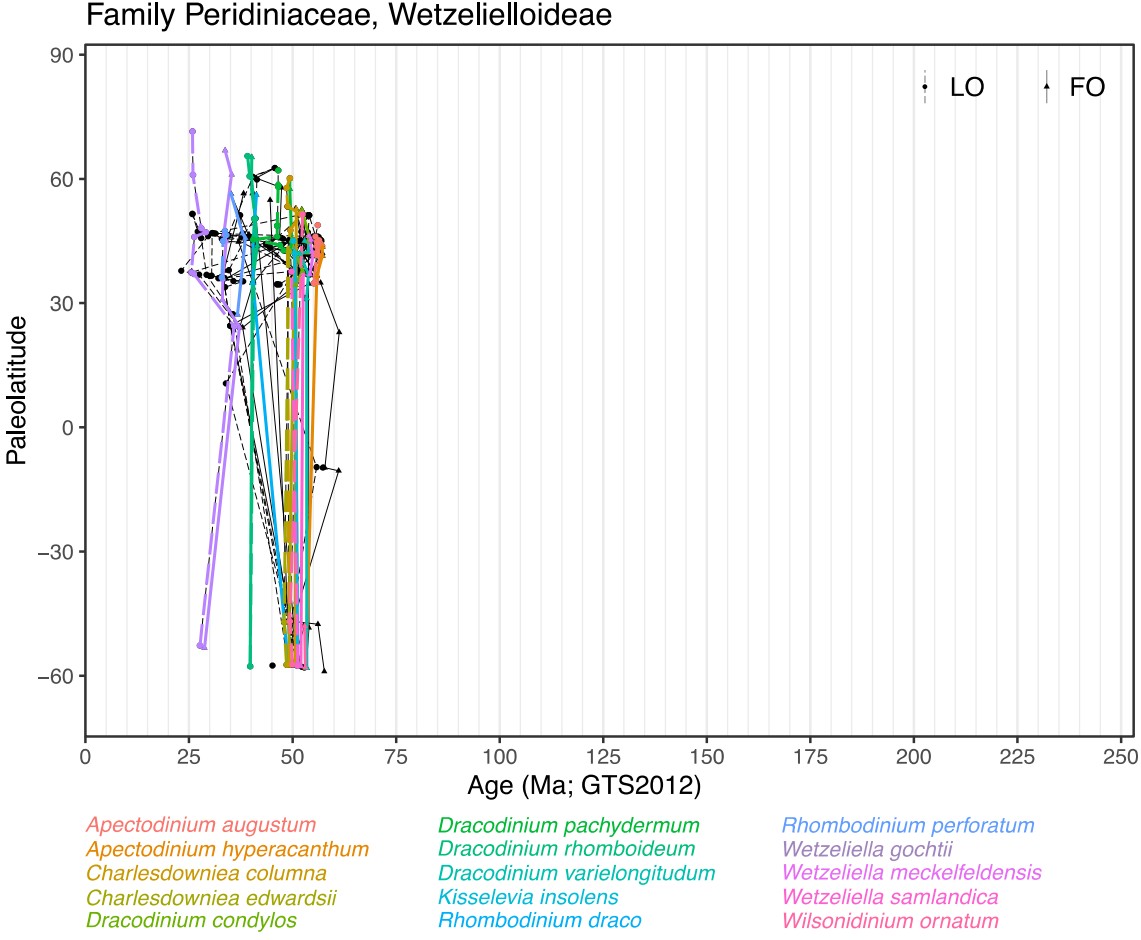


**Figure 23. As Figure 5, but for the Family Peridiniaceae, subfamily Wetzelielloideae.**

Other (Fig. 24)

*Remarks:* There is one quasi-synchronous event in this rest group: the FO of *Ovoidinium cinctum* around 129 Ma.



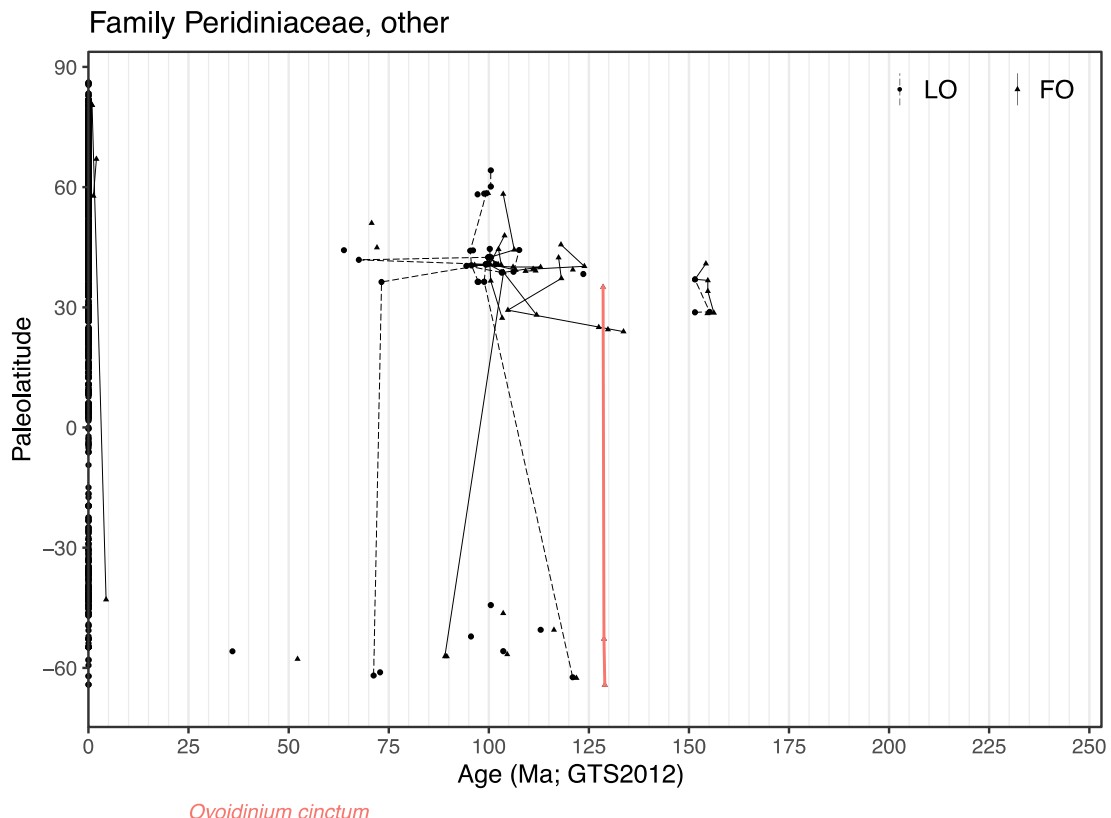

*Ovoidinium cinctum*


**Figure 24: As Figure 5, but for other subfamilies in the Family Peridiniaceae.**

Family Protoperidiniaceae (Fig. 25)

*Range:* Protoperidiniaceae first appear in the Santonian (FO of *Phelodinium magnificum*) and range into the modern with 30

species in 13 genera, which is exceptionally diverse for modern cyst families. Species have oldest first occurrences in low

latitudes than in high latitudes. Events are extremely diachronous.

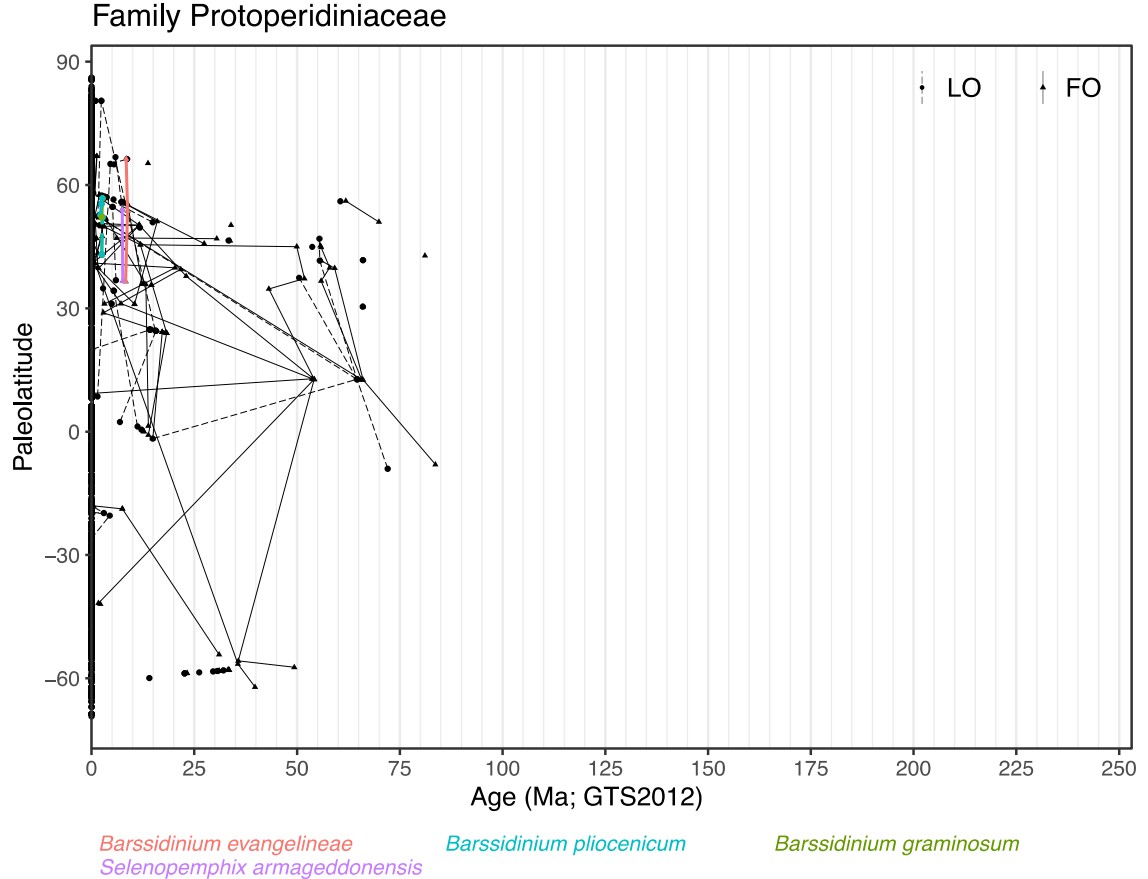

Figure 25: As Figure 5, but for the Family Protoperidiniaceae.

Order Nannoceratopsiales

Family Nannoceratopsiaceae (Fig. 26)

*Range:* Cysts from the Family Nannoceratopsiaceae occur from late Sinemurian (191 Ma, FO of *Nannoceratopsis deflandrei senex*) to the mid-Kimmeridgian (~155 Ma, LO of *Nannoceratopsis pellucida*).



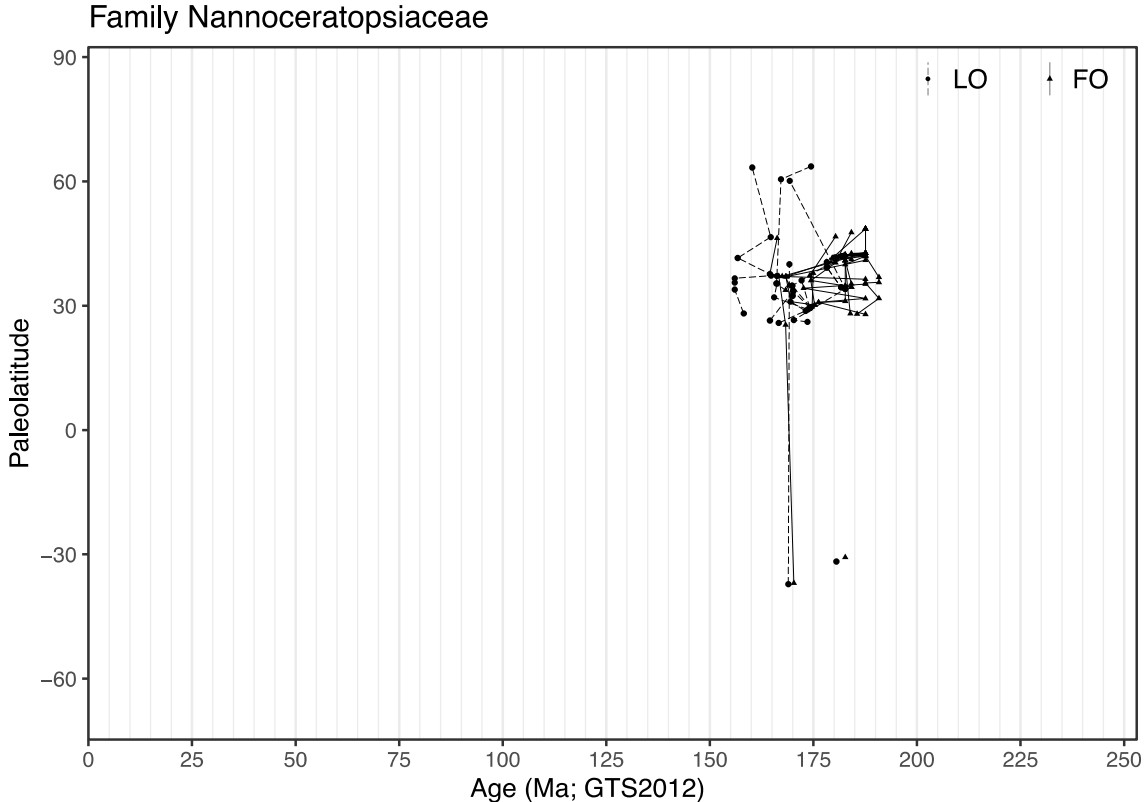


**Figure 26: As Figure 5, but for the Family Nannoceratopsiaceae.**

Order Ptychodiscales

Family Ptychodiscaceae (Fig. 27)

*Range:* This family only has entries in the late Cretaceous (91–66 Ma), where species represent fairly synchronous stratigraphic markers. Although cysts are only found in a relatively short geologic time interval, motile cells of Ptychodiscaceae are known from modern plankton.





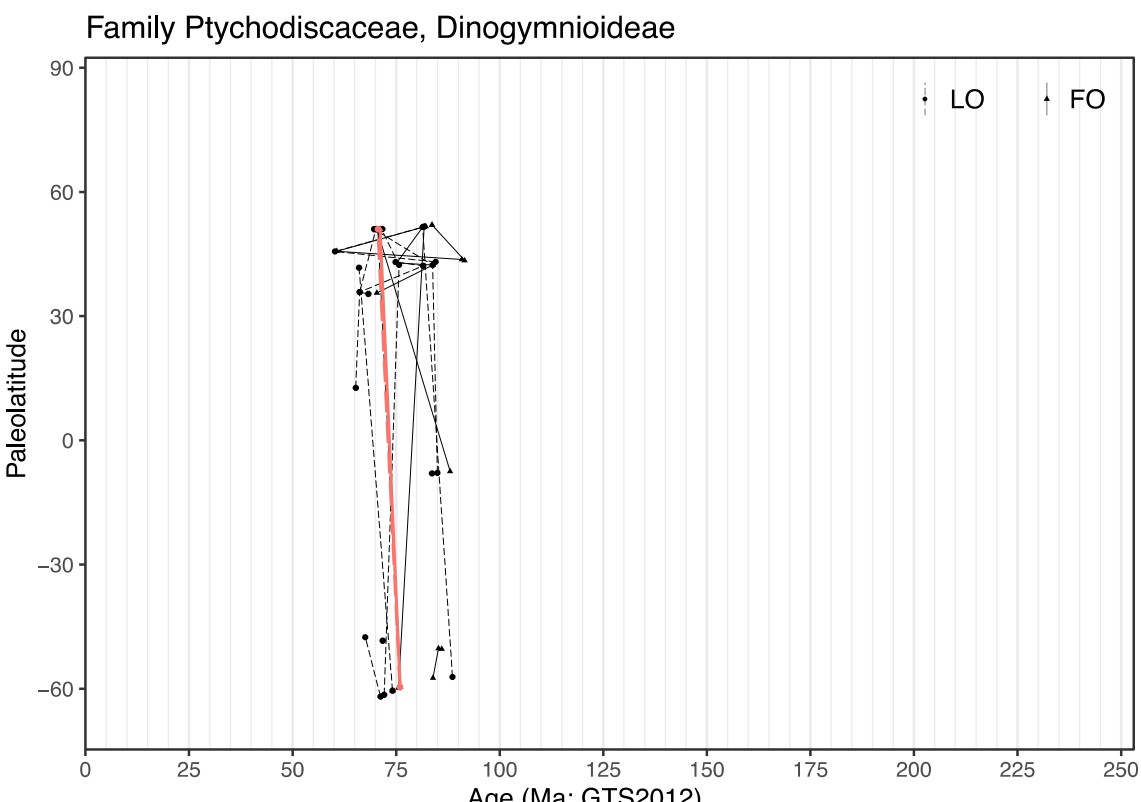

*Dinogymnium albertii*

**Figure 27: As Figure 5, but for the Family Ptychodiscaceae, subfamily Dinogymnioideae.**


Order Suessiales

Family Suessiaceae (Fig. 28)

*Range:* Suessiaceae occur in the Triassic–early Jurassic (229–182 Ma).

*Quasi-synchronous events:* LO of *Suessia swabiana*. Other events are highly diachronous.



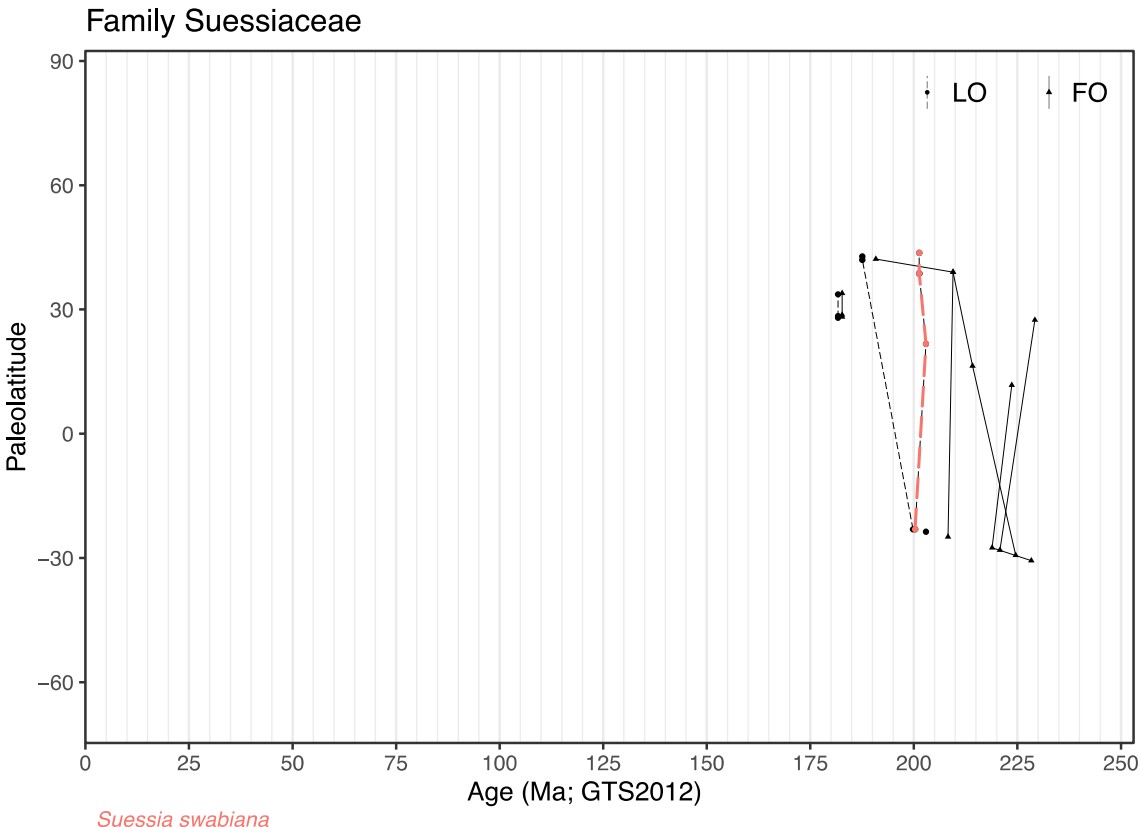


**Figure 28: As Figure 5, but for the Family Suessiaceae.**

## 4 Discussion

### 4.1 Geographic extrapolation of dinocyst events

A suite of dinocyst events throughout the entire stratigraphic record have quasi-synchronous ages across all latitudes (Fig. 5–
28). Uneven geographic spread of data, with voids in the equatorial region and the Pacific Ocean, makes global synchroneity
of these events highly uncertain. Still, the synchronous events confirm the potential and value of dinocyst biostratigraphy to
date complex sedimentary systems. It also implies that ocean connectivity did allow dinocyst species to migrate globally, as
far as their environmental tolerances permit.

Yet, the majority of dinocyst species have very diachronous ranges in DINOSTRAT, as well as latitudinally restricted
geographic spread, which confirms previous interpretations (Williams et al., 2004). With DINOSTRAT the underlying causes
of this diachroniety can now be further explored. The shortness of some of the records used in this review may lead to 'false'
events, i.e., those that represent re-appearance or temporal disappearance rather than 'true' first or last occurrences (FO and
LO, respectively). The obvious false FOs and LOs have been removed from DINOSTRAT by omitting events that occur at the





base or the top of the sections. Particularly rare species, or those occurring at the end of their preferred environmental niche,
come and go in stratigraphic sections, and these lead to 'false' events in DINOSTRAT. Although such 'false' FOs and LOs may obscure a uniform age of events over latitudes, they may still have important regional stratigraphic significance, which is why their entries are retained in DINOSTRAT. As a result, age and region of the oldest FOs and youngest LOs have the most significance for reconstruction of evolutionary patterns. Although caving of material typically falsely increases the age of oldest FOs, this is unlikely a large influence on the entries in DINOSTRAT, as most studies come from core or outcrop
material, and not from ditch cuttings, for which caving is much more likely. Reworking could falsely extend the age of youngest LOs of species. Although species that were reported as reworked in the papers have been omitted from DINOSTRAT, some reworked dinocysts could have been falsely identified as in situ in the original papers. It cannot be excluded that this causes some level of diachroniety in LOs, although this is unlikely a large factor.

The complexity of taxonomic concepts in some dinocyst genera (species definitions, or morphological continua) hinders proper
evaluation of latitudinal synchroneity of events. The reviewed literature covers 50 years, during which taxonomic concepts of dinocysts species have iteratively evolved. The extensive synonymy database of Williams et al. (2017) does deliver crucial organization of the taxonomic framework. Still, some of the subtle morphological differences in species are limited to the expert eye of individual researchers, and these may not have been recognized by others (which occasionally led to the presentation of taxa on a genus level, instead of further specification to species level). Making the taxonomic framework
consistent for all studies now included in DINOSTRAT would be a cardinal effort and will be part of the iterative setup of DINOSTRAT. For example, reviews of dinocyst taxonomic frameworks on a per-family basis, such as has been initiated for the *Spiniferites* complex (e.g., Mertens and Carbonell-Moore, 2018) could help adjusting inconsistencies in species concepts, and their stratigraphic occurrence. In any case, it must be stressed that the quality of any biostratigraphic marker is defined not only by the accuracy of the tie to the chronostratigraphic time scale, or global consistency of the age of FO or LOs, but also
by their morphological distinctiveness.

Events may also appear diachronous in DINOSTRAT because of inadequate or inaccurate tie to the chronostratigraphic time scale. In such cases, small diachroniety (~$10^{4-5}$ yr) may be related to the inherent assumption of linear sedimentation rates between age tie points. Larger diachroniety (~$10^{5-6}$ yr) may be because the zonation through which dinocyst events were calibrated to the chronostratigraphic time scale is diachronous. For calibrations against magnetostratigraphy (Tier 1, 2) this is
unlikely, and could occur only when magnetochrons were wrongly interpreted in the sites used. For events calibrated against Cenozoic nannoplankton and foraminifer zonations (in Tier 3, 4) this is also unlikely, as these events are relatively robustly calibrated to chronostratigraphy (Watkins and Raffi, 2020; Petrizzo et al., 2020). Less robust are the Mesozoic ammonite zonation schemes, which have shown to be quite diachronous themselves latitudinally (e.g., Ogg and Hinnov, 2012a, b and references therein). The geographic variability in ages of zone boundaries, but also numerous adjustments of zone definitions
throughout the past 50 years, further complicates accurate tie of dinocyst events with ammonite data to the GTS2012. So far, the majority of Mesozoic dinocyst events were calibrated against these ammonite zonations, which makes their absolute tie to



the chronostratigraphic time scale most uncertain. A major challenge for future versions of DINOSTRAT is to improve the independent age control of Mesozoic calibrated dinocyst events.

Also, ecological reasons could cause geographically diachronous events. When local environmental or depositional conditions change, assemblages adjust, which leads to local and temporal (dis)appearances of species that may be falsely interpreted as extinction or origination events. If so, dinocyst taxa associated to the most dynamic environmental niches on the continental shelf are expected to have the most diachronous events. Indeed, there are particularly diachronous events in Goniodomaceae and Protoperidinioideae – both Families are associated to near-shore depositional settings (Zonneveld et al., 2013; Sluijs et al., 2005; Frieling and Sluijs, 2018), that are most environmentally dynamic. Settings in which these species occur offshore, such as upwelling regions (Sangiorgi et al., 2018), or hyperstratified waters (Reichart et al., 2004; Cramwinckel et al., 2019), are environmentally equally dynamic. In contrast, families typically associated to offshore conditions, such as the Wetzeliellioideae (Frieling and Sluijs, 2018) reveal much more synchronous events. For regional stratigraphy, the diachroniety is of less concern because these events can still be used for regional stratigraphic correlation (e.g., as in Vieira and Jolley, 2020). It does mean that for such species, dinocyst biostratigraphy applies regionally, and caution should be taken to extrapolate event ages far outside of these regions. There are also species that clearly show regional inconsistency of origination or extinction ages as a result of climate change – e.g., *Melitasphaeridium choanophorum* had a much wider geographic distribution during warmer past climates and a progressively younger LO in lower latitudes as climate cooled (Fig. 4).

Diachroniety is usually larger between latitudinal bands than within latitudinal bands. The sparsity of records from the Southern high latitudes complicates robust assessment of interhemispheric differences in dinocyst event ages. In the Mesozoic, the diachroneity is likely related to the inadequate tie of events to the international time scale. DINOSTRAT is short of Mesozoic records that are tied to other stratigraphic tools than ammonites. For the Cenozoic, the diachroneity between hemispheres cannot be explained by inadequate calibration, since many events are calibrated against magnetostratigraphy. For those, environmental reasons must be at play. While in the early Paleogene many dinocyst events are quasi-synchronous (events within the Wetzeliellioideae, of *Cerodinium* and *Palaeoperidinium*), in the late Paleogene and Neogene diachroneity seems to become stronger. This may be in part because of stronger latitudinal temperature gradients as global average climate cools (Cramwinckel et al., 2018; Westerhold et al., 2020), which creates more diverse ecological niches and complicates latitudinal migration.

Many dinoflagellate cyst species and higher generic ranks have their oldest first occurrence and youngest last occurrence in Northern Hemisphere mid-latitudes (see, e.g., Areoligeraceae, Cladopyxiaceae, Comparodiniaceae, Goniodomaceae, Nannoceratopsiaceae, Palaeoperidinioideae, Wetzeliellioideae; Figs. 5, 7, 18, 8, 26, 22, 23). This may be because of a much higher density of records at those latitudes. However, the vast continental shelf area in Europe throughout the Mesozoic and much of the Cenozoic did likely serve as the perfect habitat for taxa to find a new niche and to linger on. A higher record density in Southern Hemisphere and equatorial regions should shed light on this idea.





**4.2 Evolutionary patterns in dinocyst (sub-) families**

DINOSTRAT presents for the first time a quantitative overview of stratigraphic and paleolatitudinal distribution of fossil and modern dinoflagellate cyst taxa. Through that, it refines with coherent, independent, open-access data the evolutionary patterns presented previously (e.g., Fensome et al., 1993; McRae et al., 1996), and adds their latitudinal distribution through time. Following up on 60 million years of experimentation in cyst-formation among a wide group of dinoflagellates (Figs. 13, 15, 16, 18–20, 26, 28), Gonyaulacoid dinocysts developed their most fundamental taxonomic features in a rapid diversity phase

in the Bajocian (~169 Ma) likely on vast continental shelf areas on the European continent (Figs. 5, 9–12, 17). The extremely high diversity in Gonyaulacoid dinocysts in the late Jurassic and Cretaceous is reflected in the density of the events in DINOSTRAT. Peridinioid dinocyst taxa strongly diversify in the late Cretaceous and Paleogene (Figs. 21–25). The decline in dinocyst diversity in the Neogene is visible in the scarcity of FOs from 25 Myrs onwards (except in Protoperidinioideae). DINOSTRAT allows to further explore spatial patterns in dinoflagellate cyst evolution in the future.

**4.3 Functionality of DINOSTRAT**

Once downloaded, DINOSTRAT can be filtered by location, allowing users to compare newly generated dinocyst chronologies to nearby calibrated regional dinocyst events. DINOSTRAT can also be filtered by species, genus or higher taxonomic rank, for further evaluation of the latitudinal spread of any species of interest. The data in DINOSTRAT is readily visualized in Supplement 2, and these plots can be adjusted and reproduced using the R markdown file "plot creator" in Bijl, 2021. The

community is invited to contact the first author either via email or through GitHub, with suggestions, error reports, and/or additional papers or data to be entered, so that the data content of DINOSTRAT is iteratively improved.

**4.4 Future directions**

DINOSTRAT will be regularly updated. Annual minor updates include addition of sites, adjustments in the current entries (e.g., through the feedback process), or minor revisions in taxonomy/stratigraphy. Major updates will occur in a 3-year cycle

and are the result of new Geologic Time Scales, or profound revisions in dinocyst taxonomic concepts. Major updates will be accompanied by a short communication in this journal, minor updates will be communicated through the GitHub repository. Updates of the Geologic Time Scale (e.g., to GTS2020 (Gradstein et al., 2020)) will be implemented once the metadata of that Geologic Time scale have become available. All versions of DINOSTRAT will remain archived on GitHub.

**5 Data availability**

The database is available under a CC-BY 4.0 license on GitHub (Bijl, 2021; https://github.com/bijlpeter83/DINOSTRAT.git; DOI:10.5281/zenodo.4471204). The database consists of 4 csv files: (1) "Paleolatitude.csv"; paleolatitude and present-day position of sites in DINOSTRAT, (2) "modernsp.csv"; the site locations of core top sediments, (3) "modernsp.csv"; a modified



modern dinocyst dataset, and (4) "Dinoevents_Jan2021.csv"; the calibrated dinocyst events. "Plot creator.Rmd" is an R markdown file to reproduce the figures presented in this paper.

## 6 Conclusions

This paper presents the database DINOSTRAT version 1.0 (Bijl, 2021), a database containing >8500 entries of regional dinoflagellate cyst first and last occurrences (events) from 1914 species, in 189 sites. Geographic distribution of sites used in DINOSTRAT is strongly concentrated in the northern hemisphere mid-latitude, notably in Europe and the North Atlantic, and few sites are in the Pacific or Southern Hemisphere. Ages of events were calibrated using their tie to the Geologic Time Scale. The paper presents the location and age of origin of modern dinoflagellate cyst species, reviews the age range and geographic spread modern and extinct dinoflagellate cyst taxa and highlights the most latitudinally synchronous dinoflagellate cyst events. Many dinocyst taxa show quasi-synchronous events latitudinally, which can be widely used to stratigraphically date complex sedimentary sequences. Latitudinal diachroneity in events can be the result of either inadequate tie to the chronostratigraphic time scale, false interpretations of 'true' events, complicated species concepts or paleoceanographic reasons. In any case, it dictates caution to extrapolate ages of dinocyst events to far distances, and demonstrates the need for regionally calibrated dinocyst zonations, which DINOSTRAT here provides. It further provides solid foundation to review spatio-temporal patterns in dinoflagellate cyst evolution, dispersal, and extinction. DINOSTRAT is freely available under CC-BY 4.0 license. It allows the user to filter by region, or by species, genus, or higher taxonomic rank.

## 7 Supplements

- Supplement 1: Table of conversions of published zones to those in GTS2012
- Supplement 2: Zip file containing ages and latitudes of events in individual dinoflagellate cyst species (1914 plots), grouped per genus (460 plots), per Family (28 plots), of modern cyst species (92 plots), and the range charts for all Sites (189 plots).

## 8 Competing interests

Author declares no conflict of interests

## 9 Acknowledgements

The LPP Foundation has financially supported the development of DINOSTRAT. I thank Henk Brinkhuis, Bas vd Schootbrugge and Appy Sluijs for useful discussions. The 'Advanced course in organic-walled dinoflagellate cyst taxonomy, stratigraphy and paleoecology' has been a great 'playground' to discuss progress in the field, and for that I have Martin Head,





600 Martin Pearce, Jörg Pross, Jim Riding, Francesca Sangiorgi and Poul Schiøler to thank. I acknowledge the then research assistants who helped building predecessors of DINOSTRAT: Tjerk Veenstra, Keechy Akkerman and Caroline van der Weijst. Thanks to Martin Schobben for help with the data analysis and visualization in R, and Douwe van Hinsbergen for help reconstructing the paleolatitudes of the sites.

605



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
