# Peer review of "DINOSTRAT: A global database of the stratigraphic and paleolatitudinal distribution of Mesozoic-Cenozoic organic-walled dinoflagellate cysts"

_Earth System Science Data, 2021_

## Referee Comment (RC1)

[revised manuscript text omitted]

Family Gonyaulacaceae, Cribroperidinioideae

| Aldorfia aldorfensis | Cordosphaeridium funiculatum | Danea californica | Hystrichokolpoma spinosum |
| Apteodinium granulatum | Cribroperidinium tenuiceras | Danea impages | Operculodinium crassum |
| Carpatella cornuta | Cribroperidinium wetzelii | Diphyes ficusoides | |

**Figure 9: As Figure 5, but for the Family Gonyaulacoideae, subfamily Cribroperidinioideae.**

Subfamily Gonyaulacoideae (Fig. 10)

305    *Range:* The subfamily of Gonyaulacoideae includes common modern cyst genera such as *Spiniferites* spp., *Achomosphaera* spp., *Impagidinium* spp., *Nematosphaeropsis* spp. and *Tectatodinium* spp. The subfamily first occurs in the Bajocian (~170 Ma), with the FO of *Gonyaulacysta* spp. and *Tubotuberella* spp.

*Quasi-synchronous events:* species of *Achomosphaera, Ataxiodinium, Callaiosphaeridium, Corrudinium, Ectosphaeropsis Hystrichodinium, Impagidinium, Spiniferites and Unipontidinium* (Fig. 10). Events of species of *Escharisphaeridia* spp.,

310    *Gonyaulacysta* spp., and *Tubotuberella* spp., range slightly longer in Northern Hemisphere high latitudes than in mid-latitudes. Many species in this subfamily are strongly diachronous.

[Figure]

**Family Gonyaulacaceae, Gonyaulacoideae**

Achomosphaera andalousiensis
Achomosphaera neptuni
Ataxiodinium confusum
Callaiosphaeridium trycherium

Corrudinium harlandii
Ectosphaeropsis burdigalensis
Escharisphaeridia pocockii
Gonyaulacysta centriconnata

Gonyaulacysta eisenackii
Gonyaulacysta jurassica
Gonyaulacysta pectinigera
Hystrichodinium furcatum

Spiniferites manumii
Spiniferites ramosus
Tubotuberella dangeardii
Unipontidinium aquaeductum

**Figure 10: As Figure 5, but for the Family Gonyaulacaceae, subfamily Gonyaulacoideae.**

315

Subfamily Leptodinioideae (Fig. 11)

*Range:* The Leptodinioideae first appear in the Aalenian (~172 Ma, FO of *Meiourogonyaulax valensii*), and includes many species events in the Bajocian and Bathonian. Although most entries are in the Jurassic and lower Cretaceous, the subfamily ranges into the late Miocene (~8 Ma, LO of *Acanthaulax miocenica*).

320 *Quasi-synchronous events:* Events in species of *Ambonosphaera*, *Areosphaeridium* (NH), *Cooksonidium, Ctenidodinium, Dichadogonyaulax, Endoscrinium, Herendeenia, Kleithriasphaeridium, Leptodinium*, *Limbodinium, Litosphaeridium*, *Rigaudella aemula*, Sirmiodiniopsis, *Stiphrosphaeridium* and *Wanaea.*

[Figure]

**Family Gonyaulacaceae, Leptodinioideae**

Ambonosphaera calloviana
Areosphaeridium diktyoplokum
Cooksonidium capricornum
Ctenidodinium continuum
Ctenidodinium cornigera
Dichadogonyaulax culmula
Endoscrinium galeritum

Herendeenia postprojecta
Kleithriasphaeridium corrugatum
Leptodinium ambiguum
Limbodinium absidatum
Litosphaeridium arundum
Litosphaeridium siphoniphorum
Rigaudella aemula

Sirmiodiniopsis orbis
Stiphrosphaeridium dictyophorum
Wanaea digitata
Wanaea fimbriata
Wanaea indotata
Wanaea thysanota

325 **Figure 11: As Figure 5, but for the Family Gonyaulacoideae, subfamily Leptodinioideae.**

Other (Fig. 12)

*Remarks:* Other species in the Family Gonyaulacaceae could not be assigned to a subfamily. Species of Barbatacysta, *Chytroeisphaeridia*, *Glossodinium*, Hemiplacophora, Nelchinopsis, Saturnodinium, *Scriniodinium*, *Sepispinula*,
330 *Stephodinium, Trichodinium* spp. have remarkably consistent events.

[Figure]

**Family Gonyaulacaceae, other**

*Barbatacysta pilosa*
*Chytroeisphaeridia chytroeides*
*Chytroeisphaeridia hyalina*
*Glossodinium dimorphum*
*Hemiplacophora semilunifera*

*Nelchinopsis kostromiensis*
*Saturnodinium pansum*
*Scriniodinium crystallinum*
*Scriniodinium dictyotum*
*Scriniodinium pharo*

*Sepispinula ancorifera*
*Stephodinium coronatum*
*Trichodinium scarburghensis*

**Figure 12: As Figure 5, but for other genera in the Family Gonyaulacoideae.**

335   Family Mancodiniaceae (Fig. 13)

*Range:* Species of Mancodiniaceae occur in sediments from the late Sinemurian (~190 Ma) to the mid-Bathonian (~167 Ma) and seem quasi-synchronous latitudinally.

[Figure]

Family Mancodiniaceae

*Mancodinium semitabulatum*

**Figure 13: As Figure 5, but for the Family Mancodiniaceae.**

Family Pareodiniaceae (Fig. 14)

*Range:* Pareodiniaceae first appear in the late Toarcian (~176 Ma, FO of *Pareodinia halosa*) and range in northern hemisphere mid-latitudes into the Cenomanian (~95 Ma, LO of *Batioladinium jaegeri*). Events of species in *Carpathodinium, Pareodinia* (both NH only), *Aprobolocysta* and *Batioladinium* appear quasi-synchronous.

[Figure]

**Family Pareodiniaceae**

*Aprobolocysta eilema*
*Batioladinium reticulatum*
*Carpathodinium predae*
*Pareodinia ceratophora*
*Pareodinia prolongata*

**Figure 14: As Figure 5, but for the Family Pareodiniaceae.**

350   Family Scriniocassiaceae (Fig. 15)

*Range:* Scriniocassiaceae range from the Pliensbachian (~187 Ma, FO of *Scriniocassis weberi*) to the Bajocian (~169Ma, LO of *Scriniocassis weberi*) and comprise of only 3 species. Events from this family are only reported from the Northern Hemisphere.

[Figure]

Family Scriniocassiaceae

*Scriniocassis priscus*  *Scriniocassis weberi*

355

**Figure 15: As Figure 5, but for the Family Scriniocassiaceae.**

Family Shublikodiniaceae (Fig. 16)

*Range:* Cysts from the Family Shublikodiniaceae occur in the late Triassic (FO of *Rhaetogonyaulax wigginsii* in the Carnian,

360 ~230 Ma) to early Jurassic (LOs of *Dapcodinium sacculus* and *Dapcodinium ovale* in the mid-Pliensbachian, 187 Ma).

*Quasi-synchronous events:* LO of *Rhaetogonyaulax rhaetica* close to the Triassic-Jurassic Boundary, LO of *Dapcodinium priscum*.

[Figure]

Family Shublikodiniaceae

*Dapcodinium priscum*      *Rhaetogonyaulax rhaetica*

**Figure 16: As Figure 5, but for the Family Shublikodiniaceae.**

Family uncertain (Fig. 17)

*Remarks*: This group of which the family is uncertain does contain several stratigraphically synchronous species (Fig. 17). Ranges of species of Amiculosphaeridia, *Atopodinium, Batiacasphaera, Cleistosphaeridium, Dingodinium, Distatodinium, Heslertonia, Labyrinthodinium, Membranilarnacia, Mendicodinium, Oligokolpoma* and *Valensiella* are quasi-synchronous.

**Order Gonyaulacales Family uncertain**

[Figure]

*Amiculosphaera umbracula*  *Dingodinium spinosum*  *Membranilarnacia tenella*
*Atopodinium polygonale*  *Distatodinium biffii*  *Mendicodinium brunneum*
*Atopodinium prostatum*  *Heterosphaeridium difficile*  *Mendicodinium microscabratum*
*Batiacasphaera compta*  *Labyrinthodinium truncatum*  *Oligokolpoma galeotti*
*Batiacasphaera minuta*  *Membranilarnacia picena*  *Valensiella ovulum*
*Cleistosphaeridium polypetellum*

[revised manuscript text omitted]

---

## Referee Comment (RC2)

[referee-annotated manuscript omitted]

---

## Author Comment (AC2)

[revised manuscript text omitted]

Family Gonyaulacaceae, Cribroperidinioideae

| | | | |
|---|---|---|---|
| *Aldorfia aldorfensis* | *Cordosphaeridium funiculatum* | *Danea californica* | *Hystrichokolpoma spinosum* |
| *Apteodinium granulatum* | *Cribroperidinium tenuiceras* | *Danea impages* | *Operculodinium crassum* |
| *Carpatella cornuta* | *Cribroperidinium wetzelii* | *Diphyes ficusoides* | |

**Figure 9: As Figure 5, but for the Family Gonyaulacoideae, subfamily Cribroperidinioideae.**

Subfamily Gonyaulacoideae (Fig. 10)

305     *Range:* The subfamily of Gonyaulacoideae includes common modern cyst genera such as *Spiniferites* spp., *Achomosphaera* spp., *Impagidinium* spp., *Nematosphaeropsis* spp. and *Tectatodinium* spp. The subfamily first occurs in the Bajocian (~170 Ma), with the FO of *Gonyaulacysta* spp. and *Tubotuberella* spp.

*Quasi-synchronous events:* species of *Achomosphaera, Ataxiodinium, Callaiosphaeridium, Corrudinium, Ectosphaeropsis Hystrichodinium, Impagidinium, Spiniferites and Unipontidinium* (Fig. 10). Events of species of *Escharisphaeridia* spp.,

310     *Gonyaulacysta* spp., and *Tubotuberella* spp., range slightly longer in Northern Hemisphere high latitudes than in mid-latitudes. Many species in this subfamily are strongly diachronous.

[Figure]

**Family Gonyaulacaceae, Gonyaulacoideae**

| | | | |
|---|---|---|---|
| *Achomosphaera andalousiensis* | *Corrudinium harlandii* | *Gonyaulacysta eisenackii* | *Spiniferites manumii* |
| *Achomosphaera neptuni* | *Ectosphaeropsis burdigalensis* | *Gonyaulacysta jurassica* | *Spiniferites ramosus* |
| *Ataxiodinium confusum* | *Escharisphaeridia pocockii* | *Gonyaulacysta pectinigera* | *Tubotuberella dangeardii* |
| *Callaiosphaeridium trycherium* | *Gonyaulacysta centriconnata* | *Hystrichodinium furcatum* | *Unipontidinium aquaeductum* |

**Figure 10: As Figure 5, but for the Family Gonyaulacaceae, subfamily Gonyaulacoideae.**

315

Subfamily Leptodinioideae (Fig. 11)

*Range:* The Leptodinioideae first appear in the Aalenian (~172 Ma, FO of *Meiourogonyaulax* ﬞ *ensii*), and includes many species events in the Bajocian and Bathonian. Although most entries are in the Jurassic and lower Cretaceous, the subfamily ranges into the late Miocene (~8 Ma, LO of *Acanthaulax miocenica*).

320 *Quasi-synchronous events:* Events in species of *Ambonosphaera*, *Areosphaeridium* (NH), *Cooksonidium, Ctenidodinium, Dichadogonyaulax, Endoscrinium, Herendeenia, Kleithriasphaeridium, Leptodinium*, *Limbodinium, Litosphaeridium*, *Rigaudella aemula*, Sirmiodiniopsis, *Stiphrosphaeridium* and *Wanaea.*

[Figure]

**Family Gonyaulacaceae, Leptodinioideae**

*Ambonosphaera calloviana*
*Areosphaeridium diktyoplokum*
*Cooksonidium capricornum*
*Ctenidodinium continuum*
*Ctenidodinium cornigera*
*Dichadogonyaulax culmula*
*Endoscrinium galeritum*

*Herendeenia postprojecta*
*Kleithriasphaeridium corrugatum*
*Leptodinium ambiguum*
*Limbodinium absidatum*
*Litosphaeridium arundum*
*Litosphaeridium siphoniphorum*
*Rigaudella aemula*

*Sirmiodiniopsis orbis*
*Stiphrosphaeridium dictyophorum*
*Wanaea digitata*
*Wanaea fimbriata*
*Wanaea indotata*
*Wanaea thysanota*

**Figure 11: As Figure 5, but for the Family Gonyaulacoideae, subfamily Leptodinioideae.**

Other (Fig. 12)

*Remarks:* Other species in the Family Gonyaulacaceae could not be assigned to a subfamily. Species of Barbatacysta, *Chytroeisphaeridia*, *Glossodinium*, Hemiplacophora, Nelchinopsis, Saturnodinium, *Scriniodinium*, *Sepispinula*, *Stephodinium, Trichodinium* spp. have remarkably consistent events.

[Figure]

**Family Gonyaulacaceae, other**

*Barbatacysta pilosa*
*Chytroeisphaeridia chytroeides*
*Chytroeisphaeridia hyalina*
*Glossodinium dimorphum*
*Hemiplacophora semilunifera*

*Nelchinopsis kostromiensis*
*Saturnodinium pansum*
*Scriniodinium crystallinum*
*Scriniodinium dictyotum*
*Scriniodinium pharo*

*Sepispinula ancorifera*
*Stephodinium coronatum*
*Trichodinium scarburghensis*

[revised manuscript text omitted]

---

## Author Comment (AC3)

[revised manuscript text omitted]

---

## Author Response (AR1)

Reply to Henrik Nøhr-Hansen

I thank Henrik for the positive review, and a thorough read-through of the manuscript. His comments refer to three technical aspects:

1. correct the citations and make consistent with the reference list.

> **Proposed change: I will carefully review the reference list and make sure it is error-free.**

2. correct use of capitals in Epoch names.

> **Proposed change: I will make this consistent to the Geologic time scale 2012 (Gradstein et al., 2012)**

3. consistent use of dinocyst or dinoflagellate cyst

> **Proposed change: I will consistently use dinoflagellate cyst in the revisions of the paper.**

Minor comments:

Line 149: define LO and FO:

> **Proposed change: I will define these acronyms**

Fig. 2: add axis legend of counts:

> **Proposed change: I will add a legend at the secondary axis**

Line 281: **Proposed change: I will add new line at "Quasi-synchronous events"**

Response to Dr Nøhr-Hansens comments in the annotated PDF were replied in that PDF file.

Review Ian Harding

The author is to be congratulated on producingd an excellent manuscript to support a innovative and novel database system that has been created (over a considerable period of time one assumes) with the help of several research assistants. The content of the article is entirely appropriate to serve as support for the publication of this unique new dataset which will prove to be of enormous use to the community working on various aspects of dinocysts, dinoflagellates and wider palynological, ecological and evolutionary subjects. I would go so far as to say that this database will be a game-changer, and represents one of those major leaps in any given subject that will be viewed in future as a watershed moment that brings objective quantification to the subject of dinocyst temporal and geographic ranges, enabling

the group to be used as an even more powerful tool for the solution of geological and ecological problems.

**Response: The author thanks the reviewer for his positive response to the manuscript and the validity of the approach.**

The author provides robust explanation and justification for the selection and inclusion of the data from individual publications, including age conversions and error limits, which renders the dataset itself of high quality. The data set would indeed appear to be usable in its current format. The length of the article is appropriate, well structured and clear (though I'd suggest a few minor changes to some of the latter headings). There is some inconsistency in the use of tenses and abbreviations which need to be ironed out before the work is accepted, but all of these are minor and easily resolved, and are highlighted in the attached marked-up version of the manuscript. There are one or two abbreviations that need to be standardised through the script - these too are noted in the attachment. The figures and tables provide visually striking representations of a variety of compilations of the data in the new database, and elegantly demonstrate the enormous potential of this type of data compilation.

**Response: The author thanks the reviewer for his positive assessment on the structure of the datasets and the manuscript. The inconsistencies the reviewer has highlighted will be adjusted in a critical re-read of the manuscript. In doing this, we will follow the annotated manuscript the reviewer has attached to his review. We also acknowledge that the reviewer sees how the chosen approach highlights both the flaws in traditional assumptions in biostratigraphy and the opportunity that proper documentation of data will bring in the future.**

There is a significant issue with the reference list, which seems to be compiled in a non-standard sequence which makes it much more difficult to search than is necessary: for example, single authored publications should appear in the list before those the author wrote with others. The list as currently given seems to be in no particular order at all, with papers by the same author not listed either that way or in date order... This must be sorted out for the sake of the reader.

**Response: In a review of the manuscript, we have adjusted the order of references in the tables and the reference list according to the standards of the journal.**

The figures accompanying the manuscript serve to powerfully illustrate the power and future potential of this database. It will provide workers in the field with an extremely innovative and robust new tool that will be used to advance our understanding of many different aspects of dinoflagellates, their cysts, geological and geographic ranges, palaeoceanographic and ecological constraints on distribution and evolution. The database may be used in ways not yet foreseen. and which will only become evident as it is utilised by the community.

**Response: The author thanks the reviewer for his positive remark.**

My main concern is when it comes to the future development of the database - at present the author seems to be taking on responsibility for doing this alone (albeit perhaps supported behind the scenes by unnamed research assistants). Such an approach (and the committment to update the database every three years) seems to put an emormous responsbility on the shoulders of utlimately a single individual. I would thus urge the author to consider this point

and the potential weaknesses that this could cause. Perhaps there may be ways in which the wider global community might be able to be involved in supporting the continual refinement and updating of the database whilst still being able to maintain its high quality (other than by simply providing new data, etc.) ?

**Response: this is indeed a large task. However, the extensive literature database of the Laboratory of Palaeobotany and Palynology gives confidence that the current database has near exhausted existing literature that meets DINOSTRATS criteria for inclusion. This means that updating mostly comprises adding new literature to the database, and updating to new Geologic time scales. The author foresees the community will help with the task of keeping DINOSTRAT up to date in various ways: notifying the first author of new publications that could be included in DINOSTRAT, help with updating DINOSTRAT in connection to the regular advanced dinocyst courses that take place once every three years, and by applying collectively a better way to present dinocyst biostratigraphic data in new literature, according to the guidelines given in this paper. This includes quantitative information on present-day locality, and quantitative tie to other chronostratigraphic tools. The author will actively seek this engagement of the community.**

Overall, I consider this to be a landmark database that has the potential to revolutionise the subject area, and the author (and his previosu research assistants!) should be heartily congratulated on producing such a database that will serve to benefit the woder community.

**Response: the author thanks the author once more for his positive review. Responses to minor comments in Prof Hardings annotated PDF were replied there.**